



# Implementation of solar UV and energetic particle precipitation within the LINOZ scheme in ICON-ART

Maryam Ramezani Ziarani[1,†], Miriam Sinnhuber[1], Thomas Reddmann[1], Bernd Funke[2], Stefan Bender[2], and Michael Prather[3]

[1]Karlsruhe Institute of Technology (KIT), Institute of Meteorology and Climate Research - Atmospheric Trace Gases and Remote Sensing (IMK-ASF), Karlsruhe, Germany
[†]Current address: Mathematical Institute for Machine Learning and Data Science at the Catholic University of Eichstätt-Ingolstadt, Ingolstadt, Germany
[2]Instituto de Astrofísica de Andalucía (CSIC) Glorieta de la Astronomía s/n, 18008 Granada, Spain
[3]Earth System Science Department, University of California, Irvine, CA USA 92697

**Correspondence:** Maryam Ramezani Ziarani (maryam.ziarani@kit.edu, maryam.ramezaniziarani@ku.de)

**Abstract.** We extended the Linearized ozone scheme -LINOZ in the ICON (ICOsahedral Nonhydrostatic) -ART (the extension for Aerosols and Reactive Trace gases) model system to include $NO_y$ formed by auroral and medium-energy electrons in the upper mesosphere and lower thermosphere, and the corresponding ozone loss, as well as changes in the rate of ozone formation

due to the variability of the solar radiation in the ultraviolet wavelength range. This extension allows us to realistically represent variable solar and geomagnetic forcing in the middle atmosphere using a very simple ozone scheme. The LINOZ scheme is computationally very cheap compared to a full middle atmosphere chemistry scheme, yet provides realistic ozone fields consistent with the stratospheric circulation and temperatures, and can thus be used in climate models instead of prescribed ozone climatologies. To include the reactive nitrogen ($NO_y$) produced by auroral and radiation belt electron precipitation in

the upper mesosphere and lower thermosphere during polar winter, the so-called energetic particle precipitation indirect effect, an upper boundary condition for $NO_y$ has been implemented into the simplified parameterization scheme of the N2O/$NO_y$ reactions. This parameterization, which uses the geomagnetic $A_p$ index, is also recommended for chemistry-climate models in the CMIP6 experiments. With this extension, the model simulates realistic „tongues" of $NO_y$ propagating downward in polar witner from the model top in the upper mesosphere into the mid-stratosphere with an amplitude that is modulated by

geomagnetic activity. We then expanded the simplified ozone description used in the model by applying LINOZ version 3. The additional ozone tendency from $NO_y$ is included by applying the corresponding terms of the version 3 of LINOZ. This $NO_y$, coupled as an additional term in the linearized ozone chemistry, led to significant ozone losses in the polar upper stratosphere in both hemispheres which is qualitatively in good agreement with ozone observations and model simulations with EPP-$NO_y$ and full stratospheric chemistry. In a subsequent step, the tabulated coefficients forming the basis of the LINOZ scheme were

provided separately for solar maximum and solar minimum conditions. These coefficients were then interpolated to ICON-ART using the F10.7 index as a proxy for daily solar spectra (UV) variability to account for solar UV forcing. This solar UV forcing in the model led to changes in ozone in the tropical and mid-latitude stratosphere consistent with observed solar signals in stratospheric ozone.





## 1   Introduction

The solar influence in the middle atmosphere involves various contributors, including the ozone response triggered by both energetic particle precipitation (EPP) and ultraviolet (UV) solar radiation (Gray et al., 2010; Matthes et al., 2017; Dhomse et al., 2022; Maycock et al., 2016). Energetic particles precipitate into the atmosphere from multiple sources: Solar protons, accelerated to energies of a few hundred MeV, are associated with huge eruptions of the solar corona; Galactic Cosmic Rays (GCRs), which include particles with energies ranging from hundreds of MeV up to GeV (Anchordoqui et al., 2003; Thoudam

et al., 2016); Auroral electrons, precipitated during magnetic reconnection in the magnetotail, having energies ranging from a few keV up to hundreds of keV; and radiation belt electrons, containing energies up to several MeV during geomagnetic storms (Giovanni et al., 2020; Sinnhuber et al., 2012). The precipitation of energetic particles into the middle atmosphere contributes to the formation of a chain of ionic reactions by ionizing and dissociating species such as $N_2$ and $O_2$, producing neutral reactive radicals such as H, OH, N, and NO (Sinnhuber et al., 2012). Both $HO_x$ (H, $HO_2$) and $NO_x$ (N, NO, $NO_2$) trigger catalytic

chemical cycles associated with the mesospheric and stratospheric ozone loss (Lary, 1997; Sinnhuber et al., 2012). $HO_x$ has a shorter atmospheric lifetime compared to $NO_x$ and exhibits a higher potential for inducing ozone loss in the mesosphere (Bates and Nicolet , 1950; Nicolet , 1975; Lary, 1997). In contrast, $NO_x$ is longer-lived and can be transported downward through the stratosphere, leading to ozone loss in the stratosphere, particularly during polar winter and spring (Rozanov et al., 2012; Randall et al., 2006).

Auroral and radiation belt electron precipitation occurs more frequently than solar proton events. These particles do not penetrate as deeply into the middle atmosphere to the lower stratosphere as high-energy solar protons associated with solar proton events do, yet they can still produce larger amounts of $NO_x$ and are the main source for $NO_x$ in the high-latitudes upper mesosphere and lower thermosphere (Sinnhuber et al., 2012). $NO_x$ variations in the mesosphere and lower thermosphere due to geomagnetic activity can be considered a proxy for electron precipitation (Kirkwood et al., 2015; Hendrickx et al., 2015;

Sinnhuber et al., 2012, 2016; Barth et al., 2002).

The distinction between the direct and indirect effects of EPP arises from where NOx is produced and its subsequent impact on ozone. When NOx is produced in the mesosphere or lower thermosphere, it does not immediately affect stratospheric ozone. Instead, it is transported downward into the stratosphere within the polar vortex before causing ozone depletion, a process known as the EPP indirect effect (EPP IE) (Randall et al., 2006; Seppälä et al., 2014). In contrast, NOx produced in

the lower mesosphere or stratosphere can cause ozone depletion directly in those regions. Although both processes ultimately involve ozone loss via NOx, we use the established terms 'direct effect' and 'indirect effect' to reflect their distinct pathways and to align with common usage in the literature.

As ozone plays an important role in radiative heating in the middle atmosphere, a realistic ozone field is essential in order to obtain a reasonable description of dynamical processes (Braesicke and Pyle, 2003, 2004). Despite numerous studies on the

impact of solar forcing on the climate system through the top-down effect, conclusive results have yet to be reached. The main reason is the limited statistics that can be obtained with resource-demanding full chemistry climate models. For such studies, a fast but realistic ozone scheme is essential to achieve a sufficient number of realizations.





The ozone loss in the stratosphere, induced by the downward transport of $NO_x$ during polar winter and spring, can lead to net radiative cooling due to the reduction in UV absorption. Conversely, during the polar night, ozone loss results in net radiative heating because of the reduction in IR emission (Sinnhuber et al., 2018). These changes subsequently alter the dynamics of the middle atmosphere, initiating a chain of dynamical shifts that contribute to top-down solar forcing during polar winter and spring. This process, driven by the EPP-$NO_x$ indirect effect, appears to impact tropospheric weather systems in the high and mid-latitudes during winter and spring (Seppälä et al., 2009; Maliniemi et al., 2014; Rozanov et al., 2012; Matthes et al., 2017).

Variable solar UV is another source of ozone variability in the stratosphere (Gray et al., 2010; Matthes et al., 2017; Dhomse et al., 2022; Maycock et al., 2016). Ozone formation is driven by photolysis of $O_2$ in the UV spectral range at wavelengths less than 220 nm, and changes in the UV flux will affect the rate of formation of ozone particularly around the tropical stratopause (Gray et al., 2010; Matthes et al., 2017). The variations of solar ultraviolet radiation depend on sunspot activity that occurs in 11-year solar cycles. During solar maximum, increased levels of UV radiation lead to higher rates of oxygen photolysis, resulting in the production of ozone (Dhomse et al., 2022; Maycock et al., 2016).

The changes in radiative heating rates induced by both direct modulation of UV radiation at the tropical stratopause and indirect modulation through ozone changes alter temperatures and dynamics of the middle atmosphere (Gray et al., 2010; Matthes et al., 2017). These radiative heating changes alter the meridional temperature gradient (Holton et al., 2004), thereby affecting the zonal wind. As a result, the changes in the zonal wind can modulate the behavior of planetary waves, penetrating further down to the earth's surface, eventually impacting the lower atmospheric circulation patterns such as the Arctic Oscillation (AO) and the North Atlantic Oscillation (NAO) (Gray et al., 2010; Matthes et al., 2017; Kodera and Kuroda, 2002).

In this paper, we describe the implementation of variable solar UV radiation and particle precipitation by applying the UBC-$NO_x$ in the simplified $NO_y$ scheme and using the $NO_y$ tendency term in the linearized ozone chemistry scheme LINOZ. This scheme is incorporated into the chemistry-climate model ICON-ART, and the impact of solar variability due to EPP and changes in solar UV radiation on $NO_y$ and ozone in the middle atmosphere is assessed using ICON-ART-LINOZ. The results are compared with observations of $NO_y$ from the Michelson Interferometer for Passive Atmospheric Sounding (MIPAS) (Fischer et al., 2008), as well as with model outputs from the ECHAM/MESSy Atmospheric Chemistry (EMAC) model (Jöckel et al., 2010), as shown in Funke et al. (2014a); Sinnhuber et al. (2018). Additionally, the solar signal in stratospheric ozone derived from satellite data is compared, as shown in Maycock et al. (2016).

The description of the ICON-ART model can be found in Sections 2.1 and 2.2, and the LINOZ is discussed in Section 2.3. The experimental setup is described in Section 3. Model developments including the upper boundary condition of $NO_y$ (UBC-$NO_y$), the inclusion of the $NO_y$-based tendency term, and the incorporation of solar UV variability, detailed in Sections 4.1, 4.2, and 4.3. The quantification of the EPP and UV impact on ozone and evaluation against MIPAS observations and the EMAC model is discussed in Sections 5.1 and 5.2.





## 2 The ICON-ART Model

### 2.1 The ICON Model Description

ICON stands for ICOsahedral Nonhydrostatic model system and has been designed by a joint development between the German Weather Service (DWD) and the Max-Planck-Institute for Meteorology (MPI-M) as a unified version of numerical weather prediction (NWP) and climate configuration (Zängl et al., 2015, 2022; Jungclaus et al., 2022).

The horizontal discretization in ICON is based on an unstructured icosahedral-triangular C grid (Staniforth and Thuburn, 2012) and it has a terrain-following vertical coordinate (Zängl et al., 2022; Leuenberger et al., 2010). Employing icosahedral-triangular C grid type is advantageous for simulating polar regions, as it eliminates the singularity issue that would otherwise be encountered when applying latitude-longitude grids (Staniforth and Thuburn, 2012).

### 2.2 The ART Extension

The extension for Aerosols and Reactive Trace Gases (ART) developed at the Karlsruhe Institute of Technology (KIT) enables the inclusion of aerosols and atmospheric chemistry into ICON (Rieger et al., 2015). The ART model extension can be incorporated into ICON for numerical weather prediction (NWP) (Rieger et al., 2015) as well as climate configuration (Schröter et al., 2018).

Tras gases are included in the model via the ART coupler in a flexible way using meta-information within XML files to meet the needs for a variety of simulations without modifying the original code (Schröter et al., 2018; Weimer , 2019). ICON-ART tracers are then transported by the ICON wind fields, and can interact with the radiative heating in ICON.

The ICON-ART consists of three different chemistry approaches as listed below:

- Simple lifetime mechanism, in which a tracer has a fixed lifetime for depletion, is considered (Rieger et al., 2015).

- linearized ozone chemistry (LINOZ) (Haenel et al., 2022).

- And, full chemistry scheme via MECCA (Weimer et al., 2021).

The first and second mechanisms can be used with simplified schemes for $NO_y$ (Diekmann , 2017). Here, we use a lifetime-based chemistry approach (below 10km) and a simplified ozone scheme based on McLinden et al. (2000) (above 10km). This differs from the standard NWP configuration of ICON-ART, which typically relies on monthly climatological ozone values provided by GEMS (Global and Regional Earth-system (Atmosphere) Monitoring using Satellite and in situ data) (Hollingsworth et al., 2008).

### 2.3 The Linearized Ozone Scheme (LINOZ) as Included in ART

For a more realistic description of ozone fields compared to the prescribed ozone climatology, we have relied on a Linearized ozone scheme -LINOZ (McLinden et al., 2000). LINOZ provides a computationally efficient alternative to a full middle atmosphere chemistry scheme, while still generating ozone fields that align well with stratospheric circulation and temperatures.





In this study, we adapted the LINOZ V3 model from Hsu and Prather (2010) to focus on the interactions between $NO_y$ and
$O_3$ under solar variable forcing. $NO_y$ is obtained using the simplified parametrization scheme of the $N_2O/NO_y$ (SIMNOY)
reactions from Olsen et al. (2001), which is the current implementation method for nitrogen chemistry in ICON-ART.

The differential equation representing the linearized ozone version 3 method follows Hsu and Prather (2010):

$$\frac{df_i}{dt} = (P - L)^0_{i\ i} + \sum_{j=1}^{j=5} \frac{\partial(P - L)_i}{\partial f_j}\bigg|_0 (f_j - f_j^0) + \frac{\partial(P - L)_i}{\partial T}\bigg|_0 (T - T^0) + \frac{\partial(P - L)_i}{\partial co_3}\bigg|_0 (co_3 - co_3^0) \tag{1}$$

For $i = 1, \ldots, 4$ and $j = 1, \ldots, 5$, where $f_1 \equiv f_{o3}$, $f_2 \equiv f_{n2o}$, $f_3 \equiv f_{noy}$, $f_4 \equiv f_{ch4}$, and $f_5 \equiv f_{h2o}$.
In this study, we rely on $i = 1$ and $3$ only, $f_1 \equiv f_{o3}$, $f_3 \equiv f_{noy}$.

The temperature is represented by T, the overhead ozone column by $co_3$, and the ozone tendency term (P-L) by P for the
production term and L for the loss term. Subscript "0" is used to indicate the partial derivative evaluated at the respective clima-
tological value, and climatological values are shown with superscript "0" (Hsu and Prather, 2010). The tabulated coefficients
have been calculated for 25 pressure altitudes, 18 latitudes, and 12 months (Hsu and Prather, 2010).
To simplify the model for our specific focus, we made the following adjustments:

- Fixed climatology for $CH_4$ and $H_2O$: While this assumption may not capture long-term variations, it allows us to focus
  on the impacts of solar variability on ozone.

- Fixed $N_2O$ distribution: We use a climatological distribution for $N_2O$, meaning that the production of $NO_y$ from $N_2O$
  is fixed. Although this setup does not account for feedback mechanisms where changes in ozone could affect the strato-
spheric $N_2O$ distribution and thus $NO_y$ production, it simplifies the model to highlight the solar-ozone interaction. $NO_y$
  produced from $N_2O$ is assumed to follow this fixed distribution.

- UBC for $NO_y$: In this study, we implement a density-prescribed Upper Boundary Condition (UBC) for $NO_y$, applied
  three model levels below the upper boundary. The top three levels are fixed in the vertical grid and, with the grid spacing
  used in this study, consistently fall within the $10^{-1}$hPa to $10^{-2}$hPa range. This approach was chosen over a flux-based
UBC for several reasons, as discussed in the following. In past experiments with the EMAC model, both flux-based
  and density-prescribed UBCs were tested. Results indicated that prescribing densities in the uppermost levels performed
  significantly better than the flux-based approach, particularly at $10^{-1}$hPa, as showed in Sinnhuber et al. (2018). Given
  the similar setup of ICON and EMAC, we expect the density-prescribed UBC to perform more reliably in our study as
  well. Secondly a flux-based approach depends on the accuracy of the vertical fluxes in the upper model levels. However,
these levels typically form a sponge layer where vertical motions are artificially dampened, leading to unrealistic vertical
  fluxes. This limitation was the primary reason the flux approach did not work well in EMAC, and we anticipate similar
  challenges with ICON. Lastly, the UBC we apply is based on MIPAS satellite observations, which scan up to 68 km
  altitude. These observations implicitly include both local production of $NO_y$ in the mesosphere (due to geomagnetic
  storms and auroral substorms) and transport of NO from the thermosphere into the mesosphere. A flux-based approach





would neglect the direct $NO_y$ production in the mesosphere, as it only accounts for the vertical transport from above. By prescribing densities in the upper model levels, we ensure that both sources—mesospheric production and thermospheric transport—are considered, just as they are in the MIPAS data.

– Adjustments for solar UV variability: The Linoz tables were recalculated for ozone to account for changes in solar UV, particularly in the $J-O_2$ photolysis rates. However, for $NO_y$ tendencies, recalculation was not performed, as they are
based on the existing parameterization, as noted previously. Only the J-NO rates were extended to cover the mesosphere, using rates taken from the EMAC model.

This work represents a proof-of-concept that studies of solar variability can be conducted using this fast, efficient model. In future studies, we plan to extend this work by implementing a full version of Linoz v3, recalculating the $NO_y$ tendencies for solar variability, and dynamically coupling $CH_4$, $H_2O$, and $N_2O$ to improve the representation of chemical and
dynamical processes under varying solar conditions.

## 3   Experimental Setup

The ICON modelling system allows for different physics parameterizations to meet the needs of a variety of applications. In this study, we focused on a model experiment using the numerical weather prediction (NWP) configuration (Rieger et al., 2015). Free-running model experiments were conducted in a transient setup from 2000 to 2010, excluding the first 2.5 years
to allow for model spinup. The simulations were performed on a global R2B4 grid which corresponds to a grid resolution of approximately 160 km, with a vertical resolution of 90 levels up to an altitude of 75 km (150 km in upper atmosphere setup), and a model time step of six minutes for the physics and chemistry calculations. Results were output on a daily basis.

Ozone was calculated using the linearized LINOZ scheme, without coupling back to the radiation scheme to ensure the same dynamical behaviour in all model experiments. Polar chemistry was activated in the simulations following Haenel et al.
(2022). The experiments utilized the following forcing and boundary conditions: sea surface temperature (SST) and sea ice concentration (SIC) were taken from Taylor et al. (2000), solar irradiation was based on Lean et al. (2005), greenhouse gases (RCP4.5) were adopted from Riahi et al. (2007), and tropospheric and stratospheric aerosols were based on (Stenchikov et al., 1998, 2004, 2009).

Three model experiment were carried out within our study: Experiment 1: without the Upper Boundary Condition of $NO_y$
(UBC-$NO_y$), constant solar miminum (BASE). Experiment 2: with variable UBC-$NO_y$, constant solar minimum (UBC-$NO_y$). Experiment 3: with variable UBC-$NO_y$, constant solar maximum (SOLMAX).





## 4 Model Developments

### 4.1 The Upper Boundary Condition of $NO_y$ (UBC-$NO_y$)

We utilized a semi-empirical model for mesospheric and stratospheric $NO_y$, as described by Funke et al. (2016) to describe the
impact of auroral and radiation belt electron precipitation on $NO_y$ in the upper mesosphere. The model is characterized by the
geomagnetic $A_p$ index.

Observations of $NO_y$ (NO, $NO_2$, $NO_3$, $HNO_3$, $HNO_4$, $ClONO_2$, and $N_2O_5$) obtained by the MIPAS Fourier transform
spectrometer on board ENVISAT between 2002 and 2012 have been used to characterize the fraction of $NO_y$ produced by
energetic particle precipitation (EPP-$NO_y$) in polar winters in both hemispheres (Funke et al., 2014a). A linear relationship
with a time lag, depending on the day of the year, latitude, and altitude, was found between EPP-$NO_y$ and the geomagnetic
$A_p$ index (Funke et al., 2014b). This relationship was used in a semi-empirical model to estimate EPP-$NO_y$ densities and their
wintertime downward transport, based on the measured global distributions of $NO_y$ compounds from 2002 to 2012 (Funke et
al., 2016).

We emphasize that the stratospheric $NO_y$ in our study is derived from both simplified parametrization scheme of the
$N_2O/NO_y$ reactions from Olsen et al. (2001) and downward transport of UBC-$NO_y$. In our simulations, NOy at model's
top without the UBC is essentially negligible. The UBC, based on MIPAS observations, provides total NOy values that include
both EPP and non-EPP components. Therefore, the difference between the reference case (without UBC-NOy) and our sim-
ulations with the UBC applied represents the additional NOy introduced through the upper boundary, which likely includes
contributions from EPP but may also contain a background of non-EPP NOy.

The transport of NOy is handled by the underlying dynamics of the ICON model, where the UBC is applied three model
levels below the top to avoid noise from the sponge layer. In these top three levels, values are overwritten by the UBC to reflect
the MIPAS-derived NOy values, while the ICON dynamics are allowed to handle transport and chemistry below this boundary.
This ensures that the model properly simulates the realistic transport of $NO_y$ through the stratosphere.

Previous experiments have shown that using volume mixing ratio (VMR) as the basis for the UBC provides more stable
results, especially in avoiding problems related to vertical wind noise. While a flux-based UBC has its own challenges, the
choice of VMR was more appropriate for this study, given the dynamics of the ICON model.

The comparison of model outputs with MIPAS data validates the model's ability to simulate the transport and chemistry of
NOy as it moves through the stratosphere. While the UBC sets the boundary at the upper altitudes, the model dynamically
alters NOy below this boundary, which is why this comparison remains valuable for understanding the impacts of $NO_y$ and
EPP within the atmosphere.

In Figure 1, we show a comparison of ICON-ART without and with UBC-$NO_y$. The inclusion of UBC-$NO_y$ leads to a
strongly enhanced $NO_y$ at the model top, particularly during polar winter, as well as a downward-propagating "tongue" of
$NO_y$ indicating transport from the upper mesosphere into the mid-stratosphere during every polar winter. Qualitatively, ICON-
ART with UBC-$NO_y$ well reproduces the known behavior of EPP-$NO_y$, with interhemispheric differences due to the differing
dynamics of the high-latitude Northern and Southern winter middle atmosphere.





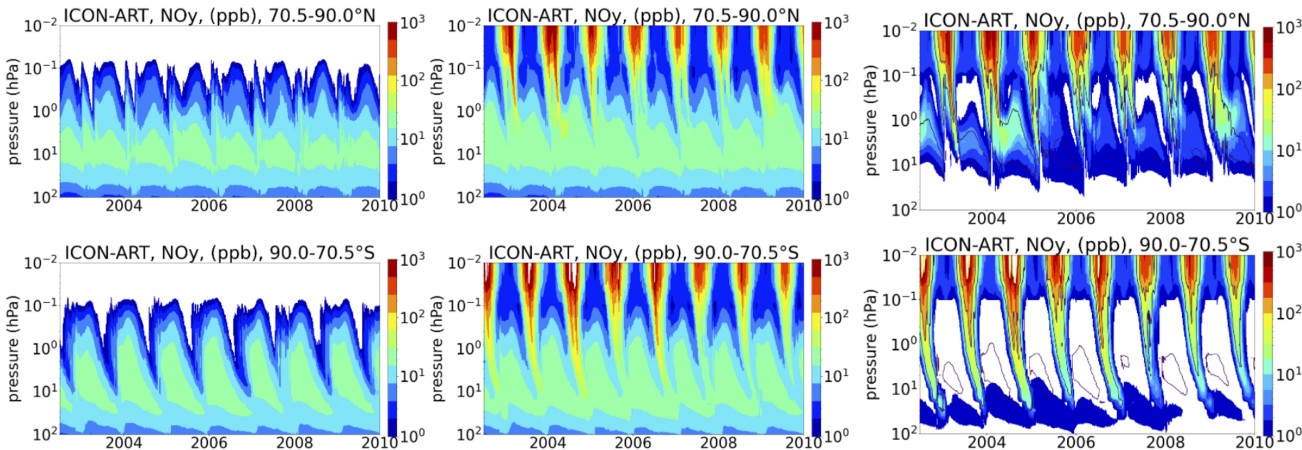

**Figure 1.** Daily mean, area-weighted NO$_y$ in 70-90°N (top) and 70-90°S (bottom) from ICON-ART. Left: experiment 1 (BASE), Middle: experiment 2 (UBC-NO$_y$), and Right: difference (UBC-NO$_y$ - BASE). Model runs are shown in 2002.5-2010 only to allow for 2.5 years of spin-up.

### 4.2 Including the NO$_y$-based Tendency Term into ICON-ART-LINOZ

In the next step of our development, we utilized LINOZ, as described in Section 2.3, to incorporate an NO$_y$-based tendency term that accounts for ozone changes in the polar stratosphere into the linearized ozone description. It is important to acknowledge that when using upper boundary NO$_y$ values, especially within the NO$_y$ tongue region, significant deviations from the climatological state occur. To enhance the reliability of the tendencies of ozone related to NO$_y$, we have re-calculated the LINOZ tables (Hsu and Prather, 2010) using a climatological NO$_y$ with upper boundary values. It's important to note that ICON is free-running, so the specific upper boundary condition used does not correspond to the model's dynamics.

### 4.3 Including the Solar UV Variation into ICON-ART-LINOZ

In addition to particle forcing, we included solar UV variability into ICON-ART to account for induced ozone changes, primarily in the tropical stratosphere. We used two spectra for November 1989 (solar maximum) and November 1994 (solar minimum) and interpolated them to the specified wavelength grid based on the photochemical box model described in Hsu and Prather (2010). Subsequently, we converted them to photon fluxes in the wavelength bins, providing two different sets of tabulated coefficients for LINOZ based on solar maximum and solar minimum conditions, following the methodology outlined in Hsu and Prather (2010).





Furthermore we calculated the values for the monthly mean 10.7 cm flux under both maximum and minimum conditions (November 1989 and November 1994) and applied a linear interpolation based on the solar activity index. F10.7 between these two states within the model.

For a better understanding of the ozone response to UV radiation, we conducted percentage difference between SOLMAX and UBC-NO$_y$ experiments relative to SOLMAX experiment. SOLMAX experiments is with the solar UV radiation fixed
at its maximum, using climatologies calculated based on the solar maximum spectrum only. As shown in Figure 2, larger ozone values, in the range of a few percent, align with observed solar signals in stratospheric ozone. Higher values at high latitudes could reflect the influence of the Brewer-Dobson circulation (Brewer, 1949) and mesospheric meridional circulation, which transport ozone from the tropical stratopause source regions to the polar mesosphere in summer and to the polar lower stratosphere in all seasons.

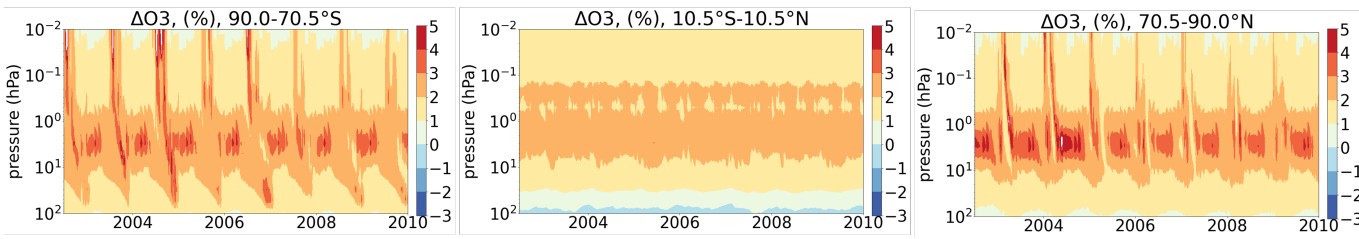

**Figure 2.** Impact of SSI on ozone in ICON-Linoz (Percentage difference between SOLMAX and UBC-NO$_y$ relative to SOLMAX). From left to right: 70-90°S, 10°S-10°N, 70-90°N respectively.

**5    Evaluation of the Particle and Solar Forcing**

**5.1    UBC-NO$_y$**

As shown in Figure 3, after the implementation of the UBC-NO$_y$, we observe a high level of qualitative agreement at the top of the atmosphere between ICON-ART and a model simulation with the EMAC model also using the UBC-NO$_y$ from Funke et al., (2016). The EMAC model employs MECCA stratospheric chemistry, specified dynamics relaxing towards ERA-interim
reanalysis data (Dee et al., 2011), and variable geomagnetic forcing for 2000-2010 (Sinnhuber et al., 2018). Despite using the same parameterization of EPP-NO$_y$, some differences between ICON-ART and EMAC NO$_y$ are apparent already at the top of the atmosphere due to differences in vertical transport and mixing.

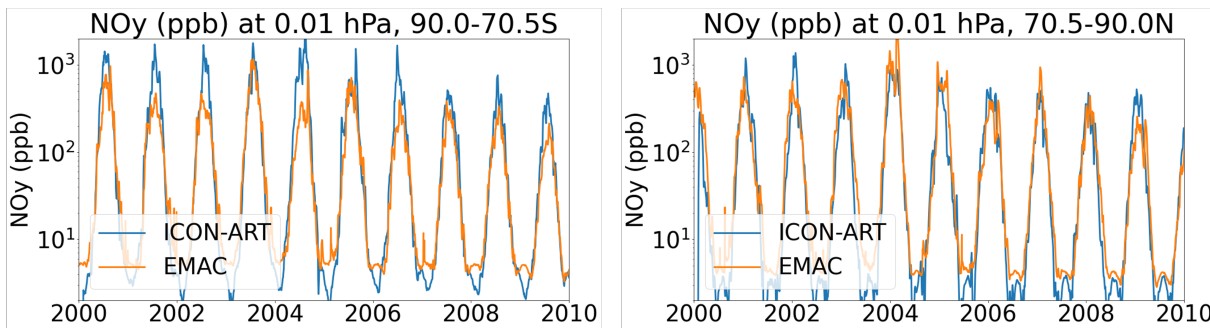

**Figure 3.** Daily mean, area weighted $NO_y$ in 0.01 hPa in 70-90°S (left) and 70-90°N (right) from the chemistry-climate model ICON-ART (experiment 2, UBC-$NO_y$) and EMAC model results from Sinnhuber et al. (2018)., also using the $NO_y$ upper boundary condition of Funke et al. (2016).

In Figure 4, $NO_y$ from ICON-ART with UBC-$NO_y$ is compared with results from the EMAC model including UBC-$NO_y$, and with MIPAS/ENVISAT v5 $NO_y$. All three data-sets reveal a significant agreement in temporal variation, vertical coverage,

and interhemispheric differences particularly in the downward propagating "tongues" of $NO_y$ during polar winters. Small differences in the year-to-year variability particularly in the Northern hemisphere are likely due to the different middle atmosphere dynamics in the free-running ICON experiments. Stratospheric $NO_y$ is gerenally higher in ICON-ART than in EMAC and MIPAS. This is even true for experiment 1 (BASE), so presumably is a feature of the SIMNOY $NO_y$ used for the stratospheric background. During the Northern Hemisphere winter of 2003/2004, $NO_y$ penetrated deeply into the stratosphere, with values

of 100 ppb around 48km/1 hPa in ICON-ART, in good agreement with EMAC and MIPAS. Due to the stronger stratospheric polar vortex in the Southern hemisphere winter, $NO_y$ is transported further down into the stratosphere there, again in good agreement between ICON-ART with UBC-$NO_y$, EMAC, and MIPAS.

In Figure 5, EPP-$NO_y$ in ICON-ART is compared to EMAC and MIPAS/ENVISAT v8. The result indicates that both models demonstrate a high degree of qualitative consistency with observations during winter. The EMAC model shows better

agreement due to its specified dynamic mode. In both models, EPP-$NO_y$ persists into summers in a very consistent way. This is not evident in the observations and could be attributed to the sensitivity cutoff related to the $NO_y$/CO correlation used to derive EPP-$NO_y$ from MIPAS/ENVISAT data.





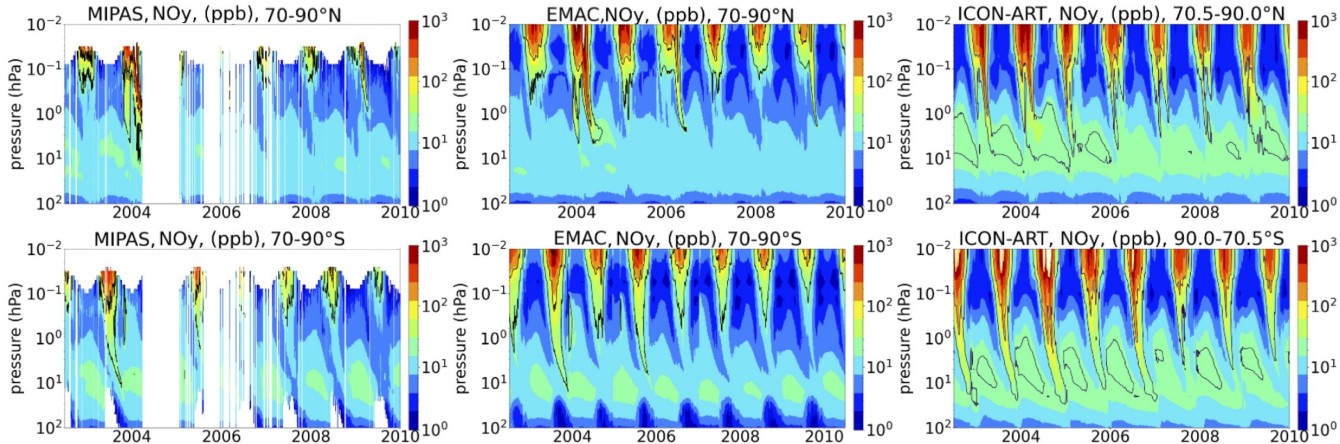

**Figure 4.** Daily mean, area-weighted NO$_y$ in 70-90°N (top) and 70-90°S (bottom) from (left) MIPAS/ENVISAT v5, (middle) EMAC, and (right) ICON-ART (UBC-NO$_y$). EMAC and MIPAS data are from Sinnhuber et al. (2018).

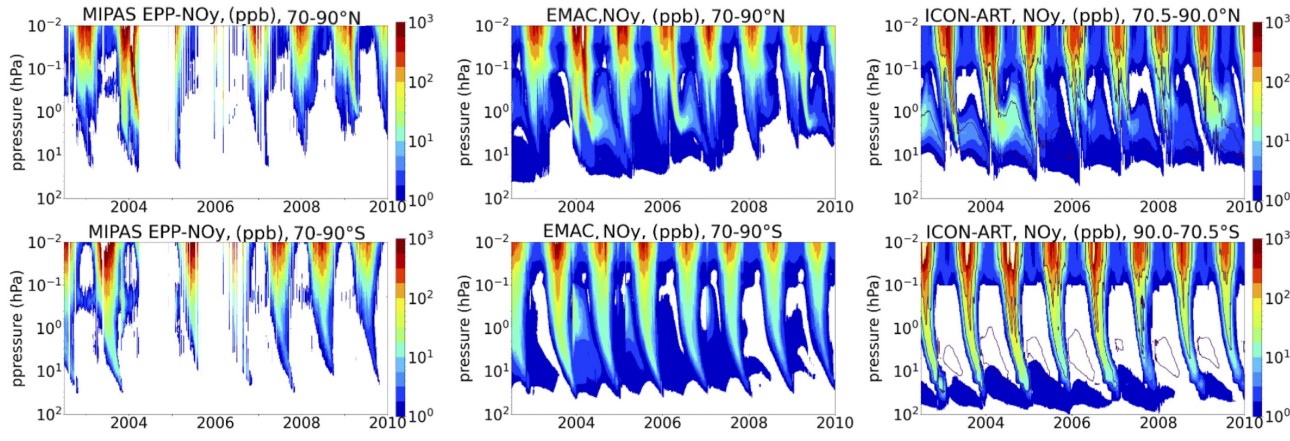

**Figure 5.** Daily mean area-weighted EPP-NO$_y$. Left: MIPAS/ENVISAT v8, courtesy (Funke et al., 2023); Middle: EMAC, difference from model run with UBCNO$_y$ to base run without UBCNO$_y$ but identical in every other respect (Sinnhuber et al., 2018); right: ICON UBCNO$_y$-BASE.

The addition of the particle forcing due to the indirect effect of EPP to the linearized ozone chemistry leds to a substantial decrease in ozone in the polar upper and mid-stratosphere in both hemispheres because of catalytic cycles that involve NO$_x$.

Figures 6 indicates the mixing ratio of the ozone fields after inclusion of the NO$_y$-based tendency in ICON-ART-LINOZ version 3 in both the Northern and Southern high latitudes compared to EMAC and MIPAS/ENVISAT v5. Comparison against



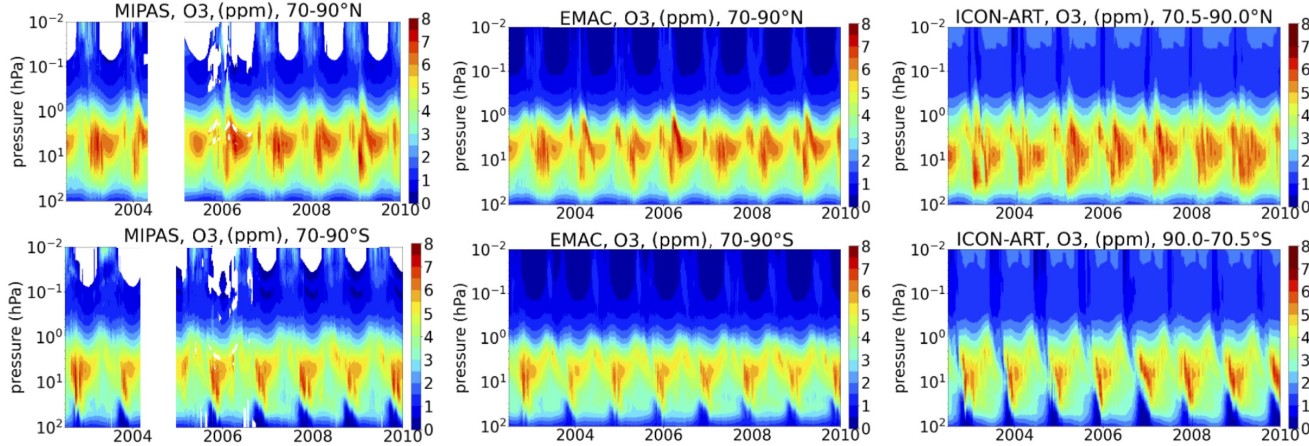

**Figure 6.** Daily mean, area weighted ozone after inclusion of the $NO_y$-based tendency in 70-90°N (top) and 70-90°S (bottom) from (left) MIPAS/ENVISAT v5, (middle) EMAC, and (right) ICON UBC-$NO_y$. The EMAC and MIPAS data are from Sinnhuber et al. (2018).

EMAC model and MIPAS/ENVISAT v5 observation shows a good agreement in the absolute values, temporal coverage of ozone change, vertical coverage and variability, as well as interhemispheric differences (Sinnhuber et al., 2018).

The pronounced simulated low ozone values in the Southern hemisphere lower stratosphere during polar winter and spring
are consistent with the Antarctic ozone hole.

Figures 7 shows the ozone change due to EPP-$NO_y$ for high Northern latitudes (70°N to 90°N) and high Southern latitudes (70°S to 90°S), for ICON-ART and EMAC. The range of values, morphology, and interhemispheric differences between the two models are consistent. The slightly larger decreases in the Southern hemisphere observed in ICON may indicate stronger downwelling and a more persistent vortex, aligning with the slightly higher EPP-$NO_y$ levels. This phenomenon is less evident
in the Northern hemisphere, which could be due to differences in the model dynamics.

Areas of low ozone develop in the mesosphere during the early winter months and descend to the mid-stratosphere by late winter/early spring in the Northern hemisphere. In the Southern hemisphere, they develop in the mesosphere during late winter/early spring and decline to the mid-stratosphere by early summer. This negative ozone response persists into the subsequent winter of 2004 around 1-10hPa of the Northern hemisphere in both models (see Figures 7). The persistent early summer ozone
depletion observed in the ICON model during 2003 may be linked to an Enhanced Stratospheric (ES) event (Manney et al., 2008) that occurred early in that year. EMAC does not show a similar ES event for 2003, while the 2006 ES event present in EMAC is not captured by ICON. These discrepancies highlight the variability in how the two models represent such events.



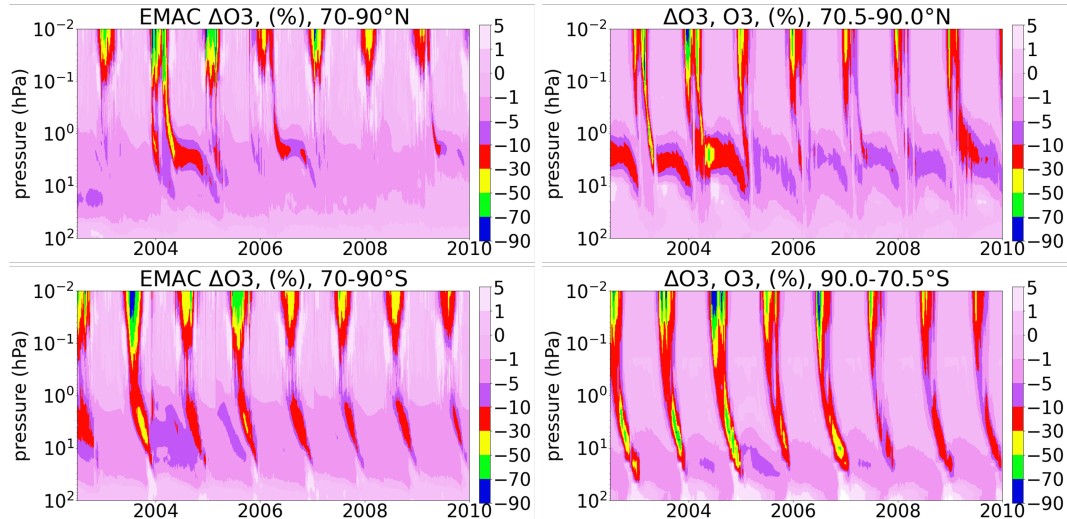

**Figure 7.** Daily mean area-weighted ozone change due to EPP-NO$_y$ in percentage in 70-90°N (top) and 70-90°S (bottom) from (left) EMAC Sinnhuber et al. (2018), and (right) ICON-ART. The contour intervalls are the same as in Sinnhuber et al. (2018) (figures 12 and 13).





## 5.2 Solar UV Variation

The impact of SSI on ozone in ICON-ART (solar maximum minus solar minimum) is shown in Figure 8. Differences of up
to 4% in the mid- and low-latitude stratosphere are observed in ICON-ART and are in good agreement with, and within,
the large spread of observations (compared, e.g., to Maycock et al. (2016), their Figures. 4 and 12). Differences in structure
could be attributed to missing radiative and dynamical feedback. At high latitudes, higher values of more than 3% are shown.
However, these cannot be compared directly against observations, as at high latitudes, the much larger changes due to particle
precipitation mask the smaller changes caused by UV variability.

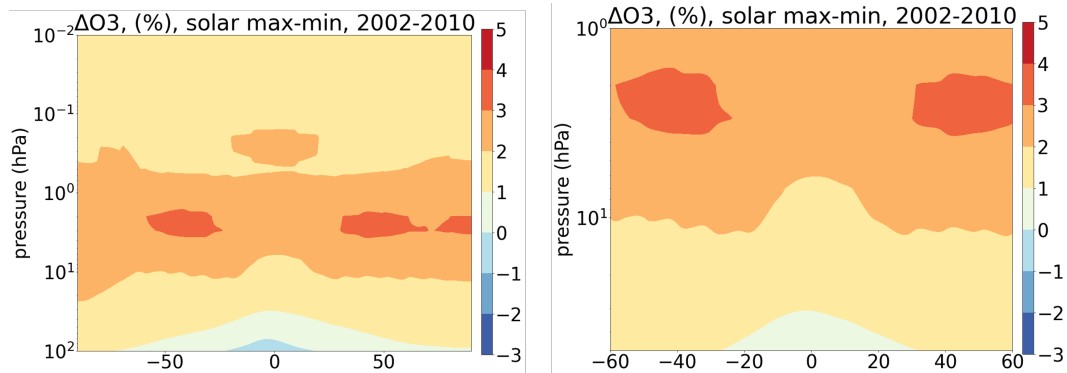

**Figure 8.** Impact of solar spectral irradiance (SSI) on ozone in ICON-ART: Percentage difference between SOLMAX and UBC-NO$_y$ relative
to UBC-NO$_y$ (left). Same, but with pressure and latitude range adapted to ozone solar signal figures (two different datasets (SAGEII/SBUV))
in Maycock et al. (2016) (right).

## 6 Conclusion

We have presented a new method of incorporating a top-down solar forcing into the stratospheric ozone, triggered by the EPP
indirect effect, by utilizing a semi-empirical model for NO$_y$ based on the geomagnetic A$_p$ index (Funke et al., 2016). This
provides a more realistic representation of the stratospheric NO$_y$ densities and its wintertime downward transport. This new
implementation of the nitrogen chemistry in ICON-ART will help improve the prediction of the ozone field in the model as a
direct response to NO$_y$.

The addition of geomagnetic forcing led to significant ozone losses in the polar upper stratosphere of both hemispheres
due to the catalytic cycles involving NO$_y$. Comparing to EMAC (Sinnhuber et al., 2018) and MIPAS (Funke et al., 2014a)
ICON-ART agrees well in the upper stratosphere (1 hPa), but it overestimates the transport into the stratosphere, leading to an
overestimation of NO$_y$ in the mid-stratosphere (at and below 10 hPa) in many (but not all) winters. The maximum ozone loss
in the mid to upper stratosphere due to the indirect effect of EPP occurs in late-winter to spring.





Considering the solar UV variability in the ICON-ART model leads to the changes in ozone in the tropical stratosphere, which is in agreement with observations (Maycock et al., 2016).

In conclusion, our study demonstrates that the inclusion of solar forcing, specifically particle precipitation and solar UV radiation, in the ICON-ART model relying on linearized ozone scheme provides realistic ozone fields.

300 **7   Acknowledgments**

We thank the BMBF project 'Solar contribution to climate change on decadal to centennial timescales' (SOLCHECK) for the financial support. The IAA team acknowledges financial support from the Agencia Estatal de Investigació́n (grant no. PID2022-141216NB-I00/AEI/10.13039/501100011033) and the Severo Ochoa grant CEX2021-001131-S funded by MCIN/AEI/10.13039/501100011033. We also thank Prof. Michael Prather for granting access to Photochemical Box-Model version 8.0.

305 **8   Code/Data availability**

The codes used in this study are available under the DOI: https://doi.org/10.35097/sh76svscb20hekj6 and are licensed under the BSD-3-Clause license. Additional details can be found in the metadata within this workspace.

**9   Competing interests**

The authors declare no competing interests.

310 **10   Author contribution**

Conceptualization, M.R.Z., M.S., T.R.; methodology, M.R.Z., M.S., T.R., B.F., S.B., M.P.; software, M.R.Z., T.R.; validation, M.R.Z., M.S., T.R.; formal analysis, M.R.Z., M.S., T.R.; investigation, M.R.Z., M.S., T.R.; writing—original draft preparation, M.R.Z.; writing—review and editing, M.R.Z., M.S., T.R., B.F., S.B., M.P.; visualization, M.R.Z., M.S.; project administration, M.S.; funding acquisition, M.S.; All authors have read and agreed to the Submitted version of the manuscript.





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
