# Peer review of "Implementation of solar UV and energetic particle precipitation within the LINOZ scheme in ICON-ART"

_Geoscientific Model Development, 2024_

## Referee Comment (RC1)

**Review of the paper**

Implementation of solar UV and energetic particle precipitation within the LINOZ scheme in ICON-ART

by
Maryam Ramezani Ziarani, Miriam Sinnhuber, Thomas Reddmann, Bernd Funke, Stefan Bender, and Michael Prather

The paper documents the implementation of an upper boundary condition for $NO_y$ (UBC-$NO_y$), and a linearized ozone parametrization (LINOZ) into the ICON-ART model. The LINOZ parametrization is further extended by terms describing ozone depletion by enhanced $NO_y$ mixing rations. The UBC-$NO_y$ allows to account for $NO_y$ enhancements during periods of high energetic particle precipitation (EPP). Besides this, the spectral solar irradiance (SSI) variability with the 11-year solar cycle is considered, by providing LINOZ coefficient tables for solar maximum and solar minimum conditions which can be scaled with the F10.7 cm solar flux. The advantage of this modelling approach is its efficiency.

**General comments**

The paper describes a very efficient way to incorporate transient, time-dependent ozone to be used in long climate projection simulations with variable SSI and EPP forcing. After some minor changes, and some additional discussions, I recommend the publication in GMD.

The introduction lacks a discussion of alternative, existing options for effectively simulating transient ozone, such as the parameterization of *Cariolle and Teyssèdre* (2007), or SWIFT (*Wohltmann et al.*, 2017; *Kreyling et al.*, 2018)

The authors should discuss the possibility of using the model system for greenhouse gas scenarios. Is it possible to simulate realistic ozone concentrations with elevated GHGs using ICON-LINOZ?

The possibility of extreme scenarios, such as the CMIP 4xCO2 scenario, which was also discussed by *Meraner et al.* (2020) in connection with the parameterization of Cariolle, should also be discussed. Does the LINOZ parameterization work for such extreme $CO_2$ scenarios?

**Specific comments**

Line 91: *. . . by a joint developement between the German Weather Service (DWD) and . . .*
Actually ICON is developed by the ICON partnership. Please replace with:
*. . . by a joint development between the German Weather Service (DWD), the Max Planck Institute for Meteorology (MPI-M), Deutsches Klimarechenzentrum (DKRZ), the Karlsruhe Institute of Technology (KIT), and the Center for Climate Systems Modeling (C2SM) . . .*

Line 95: ICON has terrain following height levels only on the lower levels. They turn into levels at constant height levels.

Line 97: Which physics parametrizations are used? Later you refer to the ICON(NWP) physics package. This information should be given here already.

Lines 124 – 125: The differential equation needs more explanations. Which variables in the equation represent the tabulated coefficients? How many coefficients are included?

Lines 137–138: *. . . applied three model levels below the upper boundary. . . .*
Reformulate:
*. . . applied to the three uppermost model levels. . . .*

Line 166: You mention an upper atmosphere setup extending to 150 km. Is the model development described in this paper also tested and available for this upper atmosphere extension of ICON? If not, you should not mention the upper atmosphere extension here, as it is confusing.

Lines 195–196: *. . . where the UBC is applied three model levels below the top to avoid noise from the sponge layer. . . .*
Please be more specific. Is the UBC applied at the three uppermost model levels, or at three levels starting at three levels below the model top? This is related to the comment to Lines 137–138.

Line 199: You describe the method to prescribe $NO_y$ volume mixing ration (VMR), here. This should already be introduced in Section 2.3 (Lines starting at 137).

Lines 220–224: Some details about the use of variable SSI are missing. For example, the number of bands used in the photochemical box model. A somewhat more detailed description of the method of *Hsu and Prather* (2010) Hsu and Prather (2010) is also missing, instead of referring to the reference.

The preparation of the SSI data is better described by the term integration rather than the term interpolation.

Line 275: You write: *. . . linked to an Enhanced Stratospheric (ES) event . . .*
Probably you meant Elevated Stratopause (ES) events.

Figure 8: Please adapt the contour level interval (0.3, or 0.5 %) of the shading, to better match the range of the data.

**Technical corrections**

Line 103: *Tras . . .* replace with: *Trace . . .*

Line 107: *. . . in which a tracer has a fixed lifetime for depletion, is considered . . .*
reformulate:
*. . . considering a tracer with a fixed lifetime for depletion . . .*

Line 109: *- And, full . . .*
replace with:
*- and a full . . .*

Line 153: *. . . Linoz . . .*
Please use consistently in the paper:
*. . . LINOZ . . .*

Line 174: *Three model experiment . . .*
Add an 's' *. . . experiments . . .*

Line 189: please change *. . . derived from both simplified parametrization scheme . . .*
to *. . . derived from both, a simplified parametrization scheme . . .*

Figure 1: You should add labels for the experiments on top of each column.

Lines 226–227: change *. . . and applied a linear interpolation based on the solar activity index. F10.7 between these two states within the model. . . .*
to *. . . and applied a linear interpolation based on the F10.7 solar activity index between these two states within the model. . . .*

Lines 228–230: reformulate these sentences: *. . . we conducted percentage difference between SOLMAX and UBC-NOy experiments relative to SOLMAX experiment. SOLMAX experiments is with the solar UV radiation fixed at its maximum, using climatologies calculated based on the solar maximum spectrum only.*
This is what I understand from the explanation. Correct, if this is a misunderstanding:
*. . . we calculated the percentage difference between the SOLMAX and UBC-NOy ex-*

*periments relative to the SOLMAX experiment. The SOLMAX experiment uses the solar UV radiation fixed at its maximum, resulting in climatologies calculated based on the solar maximum spectrum only.*

Figure 2: Caption, please change . . . *Impact of SSI on ozone* . . .
to . . . *Impact of SSI changes on ozone* . . .

Line 253: Better decribe what is shown in Figure 5. E.g.:
*In Figure 5, EPP-NOy in ICON-ART, shown as the differences between the UBC-$NO_y$ and the BASE simulations, is compared to EMAC and MIPAS/ENVISAT v8.*

Line 258: correct . . . *leds* . . . to . . . *leads* . . .

Lines 260–261: write . . . *Comparison against the EMAC model* . . .

**Bibliography**

Cariolle, D., and H. Teyssèdre (2007), A revised linear ozone photochemistry para-
meterization for use in transport and general circulation models: multi-annual
simulations, *Atmospheric Chemistry and Physics*, *7*, 2183–2196, doi:10.5194/acp-
7-2183-2007.

Hsu, J., and M. J. Prather (2010), Global long-lived chemical modes excited in a
3-d chemistry transport model: Stratospheric n2o, noy, o3 and ch 4 chemistry,
*Geophysical Research Letters*, *37*, doi:10.1029/2009GL042243.

Kreyling, D., I. Wohltmann, R. Lehmann, and M. Rex (2018), The extrapolar swift
model (version 1.0): fast stratospheric ozone chemistry for global climate models,
*Geoscientific Model Development*, *11*, 753–769, doi:10.5194/gmd-11-753-2018.

Meraner, K., S. Rast, and H. Schmidt (2020), How useful is a linear ozone para-
meterization for global climate modeling?, *Journal of Advances in Modeling Earth
Systems*, *12*, 1–19, doi:10.1029/2019MS002003.

Wohltmann, I., R. Lehmann, and M. Rex (2017), Update of the polar swift model
for polar stratospheric ozone loss (polar swift version 2), *Geoscientific Model De-
velopment*, *10*, 2671–2689, doi:10.5194/gmd-10-2671-2017.

---

## Community Comment (CC5)

Dear Reviewer, thank you very much for your positive and constructive comments. Please find our responses to your comments below, highlighted in green.

**Review of the paper**

Implementation of solar UV and energetic particle precipitation within the LINOZ scheme in ICON-ART by Maryam Ramezani Ziarani, Miriam Sinnhuber, Thomas Reddmann, Bernd Funke, Stefan Bender, and Michael Prather

The paper documents the implementation of an upper boundary condition for NOy (UBC-NOy), and a linearized ozone parametrization (LINOZ) into the ICON-ART model. The LINOZ parametrization is further extended by terms describing ozone depletion by enhanced NOy mixing rations. The UBC-NOy allows to account for NOy enhancements during periods of high energetic particle precipitation (EPP). Besides this, the spectral solar irradiance (SSI) variability with the 11-year solar cycle is considered, by providing LINOZ coefficient tables for solar maximum and solar minimum conditions which can be scaled with the F10.7 cm solar ux. The advantage of this modelling approach is its efficiency.

**General comments**

The paper describes a very efficient way to incorporate transient, time-dependent ozone to be used in long climate projection simulations with variable SSI and EPP forcing. After some minor changes, and some additional discussions, I recommend the publication in GMD.

The introduction lacks a discussion of alternative, existing options for e effectively simulating transient ozone, such as the parameterization of Cariolle and Teyssedre (2007), or SWIFT (Wohltmann et al., 2017; Kreyling et al., 2018)

The following paragraph is added to the introduction: "Several parameterizations have been developed for simulating transient ozone in chemistry-climate models. The scheme introduced by Cariolle and Teyssèdre (2007) provides a linear parameterization of ozone photochemistry, including a representation of polar ozone loss, which we also adopt in our setup. Another example is the SWIFT scheme (Wohltmann et al., 2017; Kreyling et al., 2018), which uses an efficient approach based on a fourth-order polynomial fit to full chemistry simulations. While SWIFT offers high accuracy and speed, it was originally designed for use with Lagrangian transport models, making it less directly applicable to our ICON setup. In this study, we use the LINOZ scheme, which provides a computationally efficient and dynamically consistent alternative, suitable for integration into global models that requires interactive yet fast ozone chemistry."

The authors should discuss the possibility of using the model system for greenhouse gas scenarios. Is it possible to simulate realistic ozone concentrations with elevated GHGs using ICON-LINOZ?

The following paragraph is added to the introduction: "The ICON-LINOZ is capable of simulating ozone under changing greenhouse gas (GHG) conditions. N2O and CH4 are calculated from prescribed surface flux boundary conditions, allowing their long-term

evolution to influence stratospheric ozone through interactive chemistry (Hsu and Prather, 2010). Therefore, the model can reflect the impact of elevated GHGs on ozone concentrations. If future scenarios consist of substantial shifts in these trace gases or in the background climate state, the LINOZ tables can be regenerated around a new reference state to maintain accuracy in the ozone response. This flexibility makes ICON-LINOZ suitable for studies of ozone-climate interactions under a range of future GHG pathways."

The possibility of extreme scenarios, such as the CMIP 4xCO2 scenario, which was also discussed by Meraner et al. (2020) in connection with the parameterization of Cariolle, should also be discussed. Does the LINOZ parameterization work for such extreme CO2 scenarios?

The following paragraph is added to the introduction: "The LINOZ parameterization has been shown to perform well under extreme climate scenarios, such as the CMIP  $4 \times CO_2$  case discussed by Meraner et al. (2020). In their study, both the Cariolle and LINOZ v1 schemes produced reasonable ozone responses to substantial temperature increases. Our implementation of LINOZ v3 (Hsu and Prather, 2010) builds on this by addressing a key limitation identified in Meraner et al. (2020) which is the absence of Quasi-Biennial Oscillation-related feedback on NOy due to vertical transport in LINOZ v1. In LINOZ v3, this coupling is included, allowing for more realistic simulation of ozone and NOy variability, particularly in the tropical stratosphere above 10 hPa. This confirms that the ICON-LINOZ system remain applicable and robust for studying ozone under high-CO2 scenarios."

**Specific comments**

Line 91: ...by a joint development between the German Weather Service (DWD) and ... Actually ICON is developed by the ICON partnership. Please replace with: . . .by a joint development between the German Weather Service (DWD), the Max Planck Institute for Meteorology (MPI-M), Deutsches Klimarechenzentrum (DKRZ), the Karlsruhe Institute of Technology (KIT), and the Center for Climate Systems Modeling (C2SM) ...

**Revised.**

Line 95: ICON has terrain following height levels only on the lower levels. They turn into levels at constant height levels.

Revised to: It uses a hybrid vertical coordinate system that is terrain-following near the surface and transitions to constant height levels in the upper levels.

Line 97: Which physics parametrizations are used? Later you refer to the ICON(NWP) physics package. This information should be given here already.

This sentence is added: "Our study relies on ICON(NWP) physics package."

Lines 124-125: The differential equation needs more explanations. Which variables in the equation represent the tabulated coefficients? How many coefficients are included?

**Requested information are added.**

"The tabulated coefficients used in the model include: the reference tendency term (P – L)\_i^0, and the first-order partial derivatives with respect to each variable:  $\partial(P - L)_i/\partial f_j$ ,  $\partial(P - L)_i/\partial T$ , and  $\partial(P - L)_i/\partial co_3$ . These coefficients have been calculated for 25 pressure levels, 18 latitudes, and 12 months (Hsu and Prather, 2010)."

Lines 137-138: ...applied three model levels below the upper boundary. ... Reformulate: . . .applied to the three uppermost model levels. ...

**Revised.**

Line 166: You mention an upper atmosphere setup extending to 150 km. Is the model development described in this paper also tested and available for this upper atmosphere extension of ICON? If not, you should not mention the upper atmosphere extension here, as it is confusing.

**True, It is removed.**

Lines 195-196: ...where the UBC is applied three model levels below the top to avoid noise from the sponge layer. ... Please be more specific. Is the UBC applied at the three uppermost model levels, or at three levels starting at three levels below the model top? This is related to the comment to Lines 137-138.

UBC applied at the three uppermost model levels. It is revised to avoid confusion.

Line 199: You describe the method to prescribe NOy volume mixing ration (VMR), here. This should already be introduced in Section 2.3 (Lines starting at 137).

**The paragraph is moved as suggested.**

Lines 220-224: Some details about the use of variable SSI are missing. For example, the number of bands used in the photochemical box model. A somewhat more detailed description of the method of Hsu and Prather (2010) Hsu and Prather (2010) is also missing, instead of referring to the reference.

The preparation of the SSI data is better described by the term integration rather than the term interpolation.

The paragraph is revised: "In addition to particle forcing, we included solar UV variability into ICON-ART to account for induced ozone changes, primarily in the tropical stratosphere. The photochemical box model calculating the LINOZ tables applies a solar spectrum provided in 77 spectral bins. In order to implement solar spectral variations, the LINOZ tables must be re-calculated using solar spectra representing solar maximum and solar minimum conditions. The spectra applied are based on two spectra taken during the ATLAS missions in Nov 1989 (solar maximum) and 1994 (solar minimum) and prepared as described in Kunze et al. (2020) to comply with recent measurements of the solar constant. After transferring the spectra to the 77 spectral bins of the photochemical box

model (Prather 1990), McLinden et al. 2000) (here version 8.0) we calculated two sets of tables and used them for solar maximum and solar minimum runs."

Line 275: You write: ...linked to an Enhanced Stratospheric (ES) event ... Probably you meant Elevated Stratopause (ES) events.

**Revised.**

Figure 8: Please adapt the contour level interval (0.3, or 0.5 %) of the shading, to better match the range of the data.

**Revised.**

**Technical corrections**

Line 103: Tras ... replace with: Trace ...

**Revised.**

Line 107: ...in which a tracer has a fixed lifetime for depletion, is considered ... reformulate: . . .considering a tracer with a fixed lifetime for depletion ...

**Revised.**

Line 109:- And, full ... replace with:- and a full ...

**Revised.**

Line 153: ...Linoz ... Please use consistently in the paper: . . .LINOZ ...

**Revised.**

Line 174: Three model experiment ... Add an s ... experiments ...

**Revised.**

Line 189: please change ...derived from both simplified parametrization scheme ... to ...derived from both, a simplified parametrization scheme ...

**Revised.**

Figure 1: You should add labels for the experiments on top of each column.

**Revised.**

Lines 226-227: change ...and applied a linear interpolation based on the solar activity index. F10.7 between these two states within the model. ... to ...and applied a linear interpolation based on the F10.7 solar activity index between these two states within the model. ...

**Revised.**

Lines 228-230: reformulate these sentences: ...we conducted percentage difference between SOLMAX and UBC-NOy experiments relative to SOLMAX experiment. SOLMAX experiments is with the solar UV radiation fixed at its maximum, using climatologies calculated based on the solar maximum spectrum only. This is what I understand from the explanation. Correct, if this is a misunderstanding: . . .we calculated the percentage difference between the SOLMAX and UBC-NOy experiments relative to the SOLMAX experiment. The SOLMAX experiment uses the solar UV radiation fixed at its maximum, resulting in climatologies calculated based on the solar maximum spectrum only.

It is revised to: "Figure 2 shows the impact of variable SSI as the percentage difference in ozone between solar maximum (experiment. SOLMAX) and solar minimum conditions (experiment. UBC-NOy), here relative to the results of the SOLMAX experiment.

Figure 2: Caption, please change ... Impact of SSI on ozone ... to ... Impact of SSI changes on ozone ...

**Revised.**

Line 253: Better describe what is shown in Figure 5. E.g.: In Figure 5, EPP-NOy in ICON-ART, shown as the differences between the UBCNOy and the BASE simulations, is compared to EMAC and MIPAS/ENVISAT v8.

**Revised.**

Line 258: correct ...leds ... to ...leads ...

**Revised.**

Lines 260-261: write ... Comparison against the EMAC model ...

**Revised.**

**Bibliography**

Cariolle, D., and H. Teyss`edre (2007), A revised linear ozone photochemistry para- meterization for use in transport and general circulation models: multi-annual simulations, Atmospheric Chemistry and Physics, 7, 2183–2196, doi:10.5194/acp-7-2183-2007.

Hsu, J., and M. J. Prather (2010), Global long-lived chemical modes excited in a 3-d chemistry transport model: Stratospheric n2o, noy, o3 and ch 4 chemistry, Geophysical Research Letters, 37, doi:10.1029/2009GL042243.

Kreyling, D., I. Wohltmann, R. Lehmann, and M. Rex (2018), The extrapolar swift model (version 1.0): fast stratospheric ozone chemistry for global climate models, Geoscientific Model Development, 11, 753–769, doi:10.5194/gmd-11-753-2018.

Kunze, M., Kruschke, T., Langematz, U., Sinnhuber, M., Reddmann, T., and Matthes, K. (2020), Quantifying uncertainties of climate signals in chemistry climate models related to the 11-year solar cycle – Part 1: Annual mean response in heating rates, temperature, and ozone. Atmos. Chem. and Physics, 20(11). 6991-7019. 10.5194/ acp-20-6991-2020.

McLinden, C. A., Olsen, S. C., Hannegan, B., Wild, O., Prather, M. J., and Sundet, J., Stratospheric ozone in 3-D models: A simple chemistry and the cross-tropopause flux, J. Geophys. Res.-Atmos., 105, 14653–14665 (2000), doi:10.1029/2000JD900124.

Meraner, K., S. Rast, and H. Schmidt (2020), How useful is a linear ozone para- meterization for global climate modeling?, Journal of Advances in Modeling Earth Systems, 12, 1–19, doi:10.1029/2019MS002003.

Prather, M. J., and A. H. Jaffe., Global impact of the antarc-tic ozone hole: Chemical propagation, J. Geophys. Res., 95, 3,473-3,492 (1990).

Wohltmann, I., R. Lehmann, and M. Rex (2017), Update of the polar swift model for polar stratospheric ozone loss (polar swift version 2), Geoscientific Model De- velopment, 10, 2671–2689, doi:10.5194/gmd-10-2671-2017.

---

## Author Response (AR1)

Dear Reviewer, thank you very much for your positive and constructive comments. Please find our responses to your comments below, highlighted in green.

**Review of the paper**

Implementation of solar UV and energetic particle precipitation within the LINOZ scheme in ICON-ART by Maryam Ramezani Ziarani, Miriam Sinnhuber, Thomas Reddmann, Bernd Funke, Stefan Bender, and Michael Prather

The paper documents the implementation of an upper boundary condition for NOy (UBC-NOy), and a linearized ozone parametrization (LINOZ) into the ICON-ART model. The LINOZ parametrization is further extended by terms describing ozone depletion by enhanced NOy mixing rations. The UBC-NOy allows to account for NOy enhancements during periods of high energetic particle precipitation (EPP). Besides this, the spectral solar irradiance (SSI) variability with the 11-year solar cycle is considered, by providing LINOZ coefficient tables for solar maximum and solar minimum conditions which can be scaled with the F10.7 cm solar ux. The advantage of this modelling approach is its efficiency.

**General comments**

The paper describes a very efficient way to incorporate transient, time-dependent ozone to be used in long climate projection simulations with variable SSI and EPP forcing. After some minor changes, and some additional discussions, I recommend the publication in GMD.

The introduction lacks a discussion of alternative, existing options for e effectively simulating transient ozone, such as the parameterization of Cariolle and Teyssedre (2007), or SWIFT (Wohltmann et al., 2017; Kreyling et al., 2018)

The following paragraph is added to the introduction: "Several parameterizations have been developed for simulating transient ozone in chemistry-climate models. The scheme introduced by Cariolle and Teyssèdre (2007) provides a linear parameterization of ozone photochemistry, including a representation of polar ozone loss, which we also adopt in our setup. Another example is the SWIFT scheme (Wohltmann et al., 2017; Kreyling et al., 2018), which uses an efficient approach based on a fourth-order polynomial fit to full chemistry simulations. While SWIFT offers high accuracy and speed, it was originally designed for use with Lagrangian transport models, making it less directly applicable to our ICON setup. In this study, we use the LINOZ scheme, which provides a computationally efficient and dynamically consistent alternative, suitable for integration into global models that requires interactive yet fast ozone chemistry."

The authors should discuss the possibility of using the model system for greenhouse gas scenarios. Is it possible to simulate realistic ozone concentrations with elevated GHGs using ICON-LINOZ?

The following paragraph is added to the introduction: "The ICON-LINOZ is capable of simulating ozone under changing greenhouse gas (GHG) conditions. N2O and CH4 are calculated from prescribed surface flux boundary conditions, allowing their long-term

evolution to influence stratospheric ozone through interactive chemistry (Hsu and Prather, 2010). Therefore, the model can reflect the impact of elevated GHGs on ozone concentrations. If future scenarios consist of substantial shifts in these trace gases or in the background climate state, the LINOZ tables can be regenerated around a new reference state to maintain accuracy in the ozone response. This flexibility makes ICON-LINOZ suitable for studies of ozone-climate interactions under a range of future GHG pathways."

The possibility of extreme scenarios, such as the CMIP 4xCO2 scenario, which was also discussed by Meraner et al. (2020) in connection with the parameterization of Cariolle, should also be discussed. Does the LINOZ parameterization work for such extreme CO2 scenarios?

The following paragraph is added to the introduction: "The LINOZ parameterization has been shown to perform well under extreme climate scenarios, such as the CMIP 4×CO2 case discussed by Meraner et al. (2020). In their study, both the Cariolle and LINOZ v1 schemes produced reasonable ozone responses to substantial temperature increases. Our implementation of LINOZ v3 (Hsu and Prather, 2010) builds on this by addressing a key limitation identified in Meraner et al. (2020) which is the absence of Quasi-Biennial Oscillation-related feedback on NOy due to vertical transport in LINOZ v1. In LINOZ v3, this coupling is included, allowing for more realistic simulation of ozone and NOy variability, particularly in the tropical stratosphere above 10 hPa. This confirms that the ICON-LINOZ system remain applicable and robust for studying ozone under high-CO2 scenarios."

**Specific comments**

Line 91: ...by a joint development between the German Weather Service (DWD) and ... Actually ICON is developed by the ICON partnership. Please replace with: . . .by a joint development between the German Weather Service (DWD), the Max Planck Institute for Meteorology (MPI-M), Deutsches Klimarechenzentrum (DKRZ), the Karlsruhe Institute of Technology (KIT), and the Center for Climate Systems Modeling (C2SM) ...

Revised.

Line 95: ICON has terrain following height levels only on the lower levels. They turn into levels at constant height levels.

Revised to: It uses a hybrid vertical coordinate system that is terrain-following near the surface and transitions to constant height levels in the upper levels.

Line 97: Which physics parametrizations are used? Later you refer to the ICON(NWP) physics package. This information should be given here already.

This sentence is added: "Our study relies on ICON(NWP) physics package."

Lines 124-125: The differential equation needs more explanations. Which variables in the equation represent the tabulated coefficients? How many coefficients are included?

Requested information are added.

"The tabulated coefficients used in the model include: the reference tendency term (P – L)\_i^0, and the first-order partial derivatives with respect to each variable:  $\partial(P - L)_i/\partial f_j$ ,  $\partial(P - L)_i/\partial T$ , and  $\partial(P - L)_i/\partial co_3$ . These coefficients have been calculated for 25 pressure levels, 18 latitudes, and 12 months (Hsu and Prather, 2010)."

Lines 137-138: ...applied three model levels below the upper boundary. ... Reformulate: . . .applied to the three uppermost model levels. . ..

Revised.

Line 166: You mention an upper atmosphere setup extending to 150 km. Is the model development described in this paper also tested and available for this upper atmosphere extension of ICON? If not, you should not mention the upper atmosphere extension here, as it is confusing.

True, It is removed.

Lines 195-196: ...where the UBC is applied three model levels below the top to avoid noise from the sponge layer. ... Please be more specific. Is the UBC applied at the three uppermost model levels, or at three levels starting at three levels below the model top? This is related to the comment to Lines 137-138.

UBC applied at the three uppermost model levels. It is revised to avoid confusion.

Line 199: You describe the method to prescribe NOy volume mixing ration (VMR), here. This should already be introduced in Section 2.3 (Lines starting at 137).

The paragraph is moved as suggested.

Lines 220-224: Some details about the use of variable SSI are missing. For example, the number of bands used in the photochemical box model. A somewhat more detailed description of the method of Hsu and Prather (2010) Hsu and Prather (2010) is also missing, instead of referring to the reference.

The preparation of the SSI data is better described by the term integration rather than the term interpolation.

The paragraph is revised: "In addition to particle forcing, we included solar UV variability into ICON-ART to account for induced ozone changes, primarily in the tropical stratosphere. The photochemical box model calculating the LINOZ tables applies a solar spectrum provided in 77 spectral bins. In order to implement solar spectral variations, the LINOZ tables must be re-calculated using solar spectra representing solar maximum and solar minimum conditions. The spectra applied are based on two spectra taken during the ATLAS missions in Nov 1989 (solar maximum) and 1994 (solar minimum) and prepared as described in Kunze et al. (2020) to comply with recent measurements of the solar constant. After transferring the spectra to the 77 spectral bins of the photochemical box

model (Prather 1990), McLinden et al. 2000) (here version 8.0) we calculated two sets of tables and used them for solar maximum and solar minimum runs."

Line 275: You write: ...linked to an Enhanced Stratospheric (ES) event ... Probably you meant Elevated Stratopause (ES) events.

Revised.

Figure 8: Please adapt the contour level interval (0.3, or 0.5 %) of the shading, to better match the range of the data.

Revised.

**Technical corrections**

Line 103: Tras ... replace with: Trace ...

Revised.

Line 107: ...in which a tracer has a fixed lifetime for depletion, is considered ... reformulate: . . .considering a tracer with a fixed lifetime for depletion ...

Revised.

Line 109:- And, full ... replace with:- and a full ...

Revised.

Line 153: ...Linoz ... Please use consistently in the paper: . . .LINOZ ...

Revised.

Line 174: Three model experiment ... Add an s ... experiments ...

Revised.

Line 189: please change ...derived from both simplified parametrization scheme ... to ...derived from both, a simplified parametrization scheme ...

Revised.

Figure 1: You should add labels for the experiments on top of each column.

Revised.

Lines 226-227: change ...and applied a linear interpolation based on the solar activity index. F10.7 between these two states within the model. ... to ...and applied a linear interpolation based on the F10.7 solar activity index between these two states within the model. ...

Revised.

Lines 228-230: reformulate these sentences: ...we conducted percentage difference between SOLMAX and UBC-NOy experiments relative to SOLMAX experiment. SOLMAX experiments is with the solar UV radiation fixed at its maximum, using climatologies calculated based on the solar maximum spectrum only. This is what I understand from the explanation. Correct, if this is a misunderstanding: . . .we calculated the percentage difference between the SOLMAX and UBC-NOy experiments relative to the SOLMAX experiment. The SOLMAX experiment uses the solar UV radiation fixed at its maximum, resulting in climatologies calculated based on the solar maximum spectrum only.

It is revised to: "Figure 2 shows the impact of variable SSI as the percentage difference in ozone between solar maximum (experiment. SOLMAX) and solar minimum conditions (experiment. UBC-NOy), here relative to the results of the SOLMAX experiment.

Figure 2: Caption, please change ...Impact of SSI on ozone ... to ...Impact of SSI changes on ozone ...

Revised.

Line 253: Better describe what is shown in Figure 5. E.g.: In Figure 5, EPP-NOy in ICON-ART, shown as the differences between the UBCNOy and the BASE simulations, is compared to EMAC and MIPAS/ENVISAT v8.

Revised.

Line 258: correct ...leds ... to ...leads ...

Revised.

Lines 260-261: write ... Comparison against the EMAC model ...

Revised.

**Bibliography**

Cariolle, D., and H. Teyss`edre (2007), A revised linear ozone photochemistry para- meterization for use in transport and general circulation models: multi-annual simulations, Atmospheric Chemistry and Physics, 7, 2183–2196, doi:10.5194/acp- 7-2183-2007.

Hsu, J., and M. J. Prather (2010), Global long-lived chemical modes excited in a 3-d chemistry transport model: Stratospheric n2o, noy, o3 and ch 4 chemistry, Geophysical Research Letters, 37, doi:10.1029/2009GL042243.

Kreyling, D., I. Wohltmann, R. Lehmann, and M. Rex (2018), The extrapolar swift model (version 1.0): fast stratospheric ozone chemistry for global climate models, Geoscientific Model Development, 11, 753–769, doi:10.5194/gmd-11-753-2018.

Kunze, M., Kruschke, T., Langematz, U., Sinnhuber, M., Reddmann, T., and Matthes, K. (2020), Quantifying uncertainties of climate signals in chemistry climate models related to the 11-year solar cycle – Part 1: Annual mean response in heating rates, temperature, and ozone. Atmos. Chem. and Physics, 20(11). 6991-7019. 10.5194/acp-20-6991-2020.

McLinden, C. A., Olsen, S. C., Hannegan, B., Wild, O., Prather, M. J., and Sundet, J., Stratospheric ozone in 3-D models: A simple chemistry and the cross-tropopause flux, J. Geophys. Res.-Atmos., 105, 14653–14665 (2000), doi:10.1029/2000JD900124.

Meraner, K., S. Rast, and H. Schmidt (2020), How useful is a linear ozone para- meterization for global climate modeling?, Journal of Advances in Modeling Earth Systems, 12, 1–19, doi:10.1029/2019MS002003.

Prather, M. J., and A. H. Jaffe., Global impact of the antarc-tic ozone hole: Chemical propagation, J. Geophys. Res., 95, 3,473-3,492 (1990).

Wohltmann, I., R. Lehmann, and M. Rex (2017), Update of the polar swift model for polar stratospheric ozone loss (polar swift version 2), Geoscientific Model De- velopment, 10, 2671–2689, doi:10.5194/gmd-10-2671-2017.

Dear Reviewer, thank you very much for your comments and feedback. Please find our responses to your comments below, highlighted in blue.

The manuscript describes a modification of the LINOZ chemical module to include the influence of NOy influx from the thermosphere on stratospheric ozone. The idea of using fast chemical solver is very attractive for the climate community because chemical modules usually deteriorate earth system model computational performance. I definitely support the idea, but the manuscript unfortunately does not provide complete description of the suggested LINOZ improvement. For GMD is not acceptable because it is virtually impossible to reproduce many introduced but not properly described modifications. I believe that the manuscript does not present satisfactory level of novelty and cannot recommend it for the publication. Some of manuscript problems are enlisted below.

In this paper, we show recent developments in the ICON-ART model to reproduce two aspects of the solar forcing when using the linearized ozone ICON-ART-LINOZ: 1) the enhanced ozone formation in the tropical upper stratosphere due to increased SSI during solar maximum, and 2) the EPP indirect effect due to downwelling of NOy from beyond the model top due to auroral and EUV production of NO in the thermosphere. The aim of this model development is to enable us to provide a solar cycle variability in a cost-efficient way using stratospheric ozone field which are consistent with the models' stratospheric dynamics. We feel that even though not every detail can be reproduced by this approach, it provides a big step forward in terms of providing consistent ozone fields in model experiments on the decadal to centennial scale, where more comprehensive chemistry solutions are not possible. These developments are by design not novel, but building on already existing solutions. They are already pushed to the developers branch of ICON-ART, and have been released as part of the open Icon release 04/2025. As these features are part of the open release of ICON, we feel it is important to document them, and in our understanding GMD is the right Journal for this. However, based on the detailed comments of the reviewer, it seems that a more detailed discussion of some aspects of ICON itself, and of the implementation and evaluations discussed here are necessary, and we will aim to provide these.

**Issues**

Line 40: More frequent is underestimation.

Changed the sentence in line 40 to "Electron precipitation from the magnetosphere – from the auroral and radiation belt regions – occurs nearly continuously, much more frequent than solar proton events."

Line 47: Via chemical reactions, but how about dynamical processes (Seppala et al., 2025, https://doi.org/10.1038/s41467-025-55966-z)?

Thanks for pointing this out. The paper was not published when we submitted our manuscript in December 2024. Since it is now published, we will add a discussion of their results to the introduction, at the end of the paragraph discussing the direct versus indirect effects (line 52). "A recent publication by Seppala et al (2025) indicates that a direct effect on atmospheric dynamics via mesospheric HOx production and ozone loss by precipitating magnetospheric electrons in early winter might be possible as well." This is a very exciting result which could potentially resolve a discrepancy in the timing of observed and modeled responses of surface temperatures to strong geomagnetic activity. However, as this needs precipitation of very high-energy electrons to altitudes where water vapor is sufficiently abundant to form HOx (typically below 80 km), this also must be more rare and restricted to high geomagnetic activity periods. It will not act on the auroral component which is produced in the lower thermosphere nearly continuously, and which forms the main bulk of the indirect effect captured by UBCNOy. However, if the mechanism proposed by Seppala et al (2025) proves robust by future followup experiments of other chemistry-climate models, it should of course be implemented in Linoz in the future. We will add a sentence to the discussion stating that "Depending on the robustness of the pathway discussed by Seppala et al., 2025 based on follow-up model experiments, the direct impact of mesospheric ozone loss by HOx production due to precipitation from the radiation belts can be implemented in future."

**Section 2.1: Too short. What about physical processes and so on...**

We will add the following paragraph:

In the ICON model, physical processes are considered by parameterization schemes that are distinct from the dynamical core which solves the governing equations of atmospheric motion. The NWP physics package, as detailed by Zängl et al. (2015) consists of parameterizations for radiative transfer, cloud microphysics, convection, turbulent diffusion, and surface interactions. These schemes are specifically optimized for numerical weather prediction applications, which differs from the ECHAM6-based approaches used in climate modeling (Stevens et al., 2013; Jungclaus et al. 2022). ICON physics-dynamics coupling scheme distinguishes between fast processes, such as saturation adjustment and turbulence, which are calculated at shorter time steps, and slower processes, like radiation and convection, which are computed at longer intervals (Zängl et al., 2015, 2022).

Line 103: "ART coupler in a flexible way using meta-information within XML files". What does it mean?

We re-arange this sentence: "Trace gases are included in ICON via the ART coupler without modifying the original (ICON) code. A number of different mechanisms for the description of atmospheric trace gases are available with varying complexity depending on the purpose of the simulation (Schröter et al. 2018; Weimer, 2019)."

We revised the whole session:

**2.2 Chemistry and Transport in ICON-ART**

The extension for Aerosols and Reactive Trace Gases (ART) developed at the Karlsruhe Institute of Technology (KIT) enables the inclusion of aerosols and atmospheric chemistry into ICON (Rieger et al., 2015). The ART model extension can be incorporated into ICON for numerical weather prediction (NWP) (Rieger et al., 2015) as well as climate configuration (Schröter et al., 2018).

Trace gases are included in ICON-ART with the ART coupler without changing the original ICON code. This setup allows for a flexible description of atmospheric trace gases using meta information within XML files, enabling a variety of simulations with different complexities (Schröter et al., 2018; Weimer, 2019).

**2.2.1 Transport of Trace Gases**

Trace gases in ICON-ART are transported using the same nonhydrostatic dynamical core as the rest of the model, applying a finite-volume approach on an icosahedral grid (Zängl et al., 2015). Advection of tracers is taken into account using a flux-form semi-Lagrangian method, which is mass-conserving and suitable for global-scale simulations (Reinert, 2020). In addition to advective transport, ICON-ART accounts for vertical diffusion in the planetary boundary layer, where turbulent mixing is parameterized following the prognostic turbulence kinetic energy (TKE) scheme developed by (Raschendorfer (2001)).

**2.2.2 Photolysis Rates**

Photolysis rates in ICON-ART are handled differently depending on the chemistry scheme used:

- LINOZ: This scheme uses precomputed photolysis rates stored in tabulated form, calculated using the PRATMO (Prather's Atmospheric Model) code (Hsu and Prather, 2009, 2010). These rates cover the stratosphere (10-60 km) include Rayleigh scattering, and are calculated with a fixed albedo of 0.30 to account for average cloud cover. LINOZ does not calculate photolysis rates interactively; it uses these precomputed values for efficiency. It is

important to note that LINOZ does not account for J-O2 photolysis above 60 km, and Lyman-alpha photolysis of J-H2O is not included below 70 km, where its impact is minimal.

- MECCA: The full chemistry scheme (MECCA) calculates photolysis rates using CloudJ7.3 (Prather, 2015), a module that provides accurate photolysis rates based on the solar zenith angle, cloud cover, and atmospheric composition. This module is configurable and allows for accurate photolysis calculations across various atmospheric layers.

**2.2.3 Chemistry Schemes ICON-ART supports three chemistry approaches:**

- Simple Lifetime Mechanism: For tracers with a fixed e-fold decay time, providing computational efficiency without complex chemical interactions (Rieger et al., 2015).
- LINOZ: A linearized ozone chemistry scheme (McLinden et al., 2000; Hsu and Prather, 2009, 2010), optimized for the stratosphere, where solar UV and EPP impact ozone.
- MECCA: A comprehensive full chemistry scheme (Sander et al., 2011), with numerical integration managed using the Kinetic PreProcessor (KPP) (Sandu and Sander, 2006), generating Fortran90 code for solving the differential equations of the chemical mechanism. The Rosenbrock solver of the third order (Sandu et al., 1997) is used for numerical stability. For the MECCA scheme, species can be calculated individually or conceptually grouped (e.g., NOy, HOx) in order to simplify chemical interactions. However, this is not automatic. Instead, each species is calculated individually, unless explicitly defined as a group in the chemical mechanism (Sander et al., 2011). A specific example of this is the "generic RO2" approach in MECCA, where multiple organic peroxy radicals are shown by a single generic RO2 species, reducing computational cost while maintaining chemical accuracy. The MECCA setup in ICON-ART is configured using an XML file, allowing users to define or extend chemical mechanisms without modifying the model code (Schröter et al., 2018).

Section 2.2: Too short. What about transport, photolysis rates and so on? We revised the whole session, please see it above.

Lines 126-128: The LINOZ description is too short. How are P and L terms calculated? What chemical species are used for this?

The original Linoz v1 (McLinden et al., 2000) assumed climatological (monthly zonal mean) patterns for NOy, Cly, Bry, CH4, and H2O. Linoz v2

(Hsu and Prather 2009) was an updated chemical model and further tuned the activation temperatures used for the Antarctic ozone hole chemistry. Linoz v3, see Hsu and Prather, 2010, particulary the Auxiliary Material at AGU: 2009gl042243) calculates stratospheric chemistry controlling N2O, NOy, CH4, H2O, and O3, still requiring a climatology for Cly and Bry. The performance of Linoz v3 in terms of N2O, NOy, O3 and the ozone hole has been shown to be quite good with Linoz:

Ruiz, D. J. and M.J. Prather (2022) From the middle stratosphere to the surface, using nitrous oxide to constrain the stratosphere–troposphere exchange of ozone, Atmos. Chem. Phys., 22, 2079–2093, doi: 10.5194/acp-22-2079-2022.

Prather, M.J., J. Hsu, N.M. DeLuca, C.H. Jackman, L.D. Oman, A.R. Douglass, E.L. Fleming, S.E. Strahan, S.D. Steenrod, O.A. Søvde, I.S.A. Isaksen, L. Froidevaux, and B. Funke (2015) Measuring and modeling the lifetime of nitrous oxide including its variability, J. Geophys. Res. Atmos., 120, 5693–5705. doi: 10.1002/2015JD023267.

In this study, we use an O3-NOy only version of the LINOZ v3 scheme. The P (production) and L (loss) terms in this scheme are calculated using a linearized approach, where the net chemical tendency for each species is represented as a first-order Taylor expansion around climatological mean states. Specifically:

P (Production) and L (Loss) calculations:

These terms represent the net photochemical production and loss of each species.

They are calculated using precomputed coefficients that describe the sensitivity of production and loss rates to the concentrations of the relevant species, temperature (T), and overhead ozone column (CO3).

The coefficients are derived using the PRATMO photochemical box model (Hsu and Prather, 2010), which simulates the stratospheric chemistry of  $O_3$ , NOy,  $N_2O$ ,  $CH_4$ , and  $H_2O$ .

In our  $O_3$ -NOy-only setup, these coefficients are simplified to only account for the interactions between  $O_3$  and NOy, while the other species ( $N_2O$ ,  $CH_4$ ,  $H_2O$ ) are treated as fixed climatologies.

Chemical Species Involved:

In our setup, only O₃ and NOy are dynamically calculated using the LINOZ scheme.

N₂O, CH₄, and H₂O are treated as fixed climatological fields which means their concentrations do not respond to solar variability or photochemical processes.

This setup allows for efficient calculation of  $O_3$  and NOy, but it cannot capture the full solar-ozone interaction because  $N_2O$  is not dynamically calculated.

**Tabulated Coefficients:**

The coefficients for the production (P) and loss (L) terms are precomputed for 25 pressure levels ( $\sim$ 10–58 km), 18 latitudes, and 12 months (monthly climatology). These coefficients are stored in lookup tables, which are used by the model to efficiently calculate the chemical tendencies for O3 and NOy (Hsu and Prather, 2010).

Line 131: Fixed H2O means that the impacts of solar variability on HOx production and, therefore, on ozone is missing. For the mesosphere it is a serious flaw.

While ICON extends to the mesosphere, we aim at ozone in the stratosphere and stratopause region, where it is most relevant for radiative heating. We feel that while this approach might not capture the variability of mesospheric ozone in all detail, it is a huge improvement compared to many climate models that rely on fixed ozone climatologies, as it allows us to provide ozone fields consistent with the model's dynamics.

 $H_2O$  is fixed in our setup, but this is only a problem for the conversion of  $H_2O$  to  $H_2$  well above 64 km, beyond the limit of LINOZ. Our focus is on simulating stratospheric ozone and stratopause ozone, where the direct impact of HOx is limited, and the NOy-driven chemistry is the dominant factor. If solar variability is included in the LINOZ v3 tables, then the HOx response can be captured. However our setup is a simplified LINOZ (O3-NOy only) configuration, not the full LINOZ v3. This means that only O3 and NOy are calculated dynamically, while  $N_2O$ ,  $CH_4$ , and  $H_2O$  are fixed climatological fields.

Line 135: "it simplifies the model to highlight the solar-ozone interaction". It simplifies for sure but does not highlight solar-ozone interaction due to absence of N2O response to O(1D) and photolysis.

We agree with the reviewer that our current setup does not fully capture the solar—ozone interaction. This is primarily because  $N_2$ O is prescribed as a fixed climatological field and is not calculated interactively. Therefore, solar-driven variability in  $N_2$ O (via photolysis or  $O(^1D)$ ) is not shown. However, we have implemented solar UV variability in the LINOZ scheme used in ICON-ART. Following the methodology of Hsu and Prather (2010), we recalculated the photolysis rate coefficients for solar maximum (Nov 1989) and minimum (Nov 1994) conditions. These spectra were converted to photon fluxes and used to produce two sets of LINOZ tables. During simulations, solar-cycle

variability was accounted for by interpolating the coefficients based on monthly mean F10.7 values. This allows us to capture the direct influence of solar UV variability on ozone photochemistry in the stratosphere. NOy is calculated using the LINOZ v3 formulation, including photochemical production based on fixed  $N_2$ O, stratospheric and mesospheric losses, a tropospheric sink, and an upper boundary condition (UBC) that includes EPP-NOy input. Thus, while the model includes solar UV variability in ozone photolysis, the full stratospheric solar-ozone coupling, particularly via solar-driven changes in  $N_2$ O and  $CH_4$ is not represented due to their fixed climatologies. A fully interactive approach would require dynamic  $N_2$ O,  $CH_4$ , and  $H_2$ O fields, as in the complete LINOZ v3 scheme, with photochemistry recomputed for multiple solar activity levels.

**Line 139: What is the model top?**

Linoz chemistry is only good to 0.1 hPa – extending it to 0.01 hPa provides a safe UBC, but does not include the chemistry above 70 km where Ly-alpha is important and solar-cycle changes are larger. The top of ICON is at 80 km.

Line 144: If the vertical transport is wrong, prescription of the NOy density will not help, because the downward transport from the upper layers is wrong.

That is true for both approaches, and this is why it is important to compare against observations (see Fig. 1). Therefore we agree that if the vertical transport is incorrect, simply prescribing NOy densities at the upper boundary will not solve the issue. Therefore we don't prescribe NOy only in the uppermost model layer. Instead, we apply the upper boundary condition (UBC) for NOy across the top few model levels, specifically three levels below the model top. This approach ensures that the NOy densities are set below the sponge layer, where vertical motions are artificially dampened. This method allows the model's internal vertical transport to properly control the downward propagation of NOy and providing a more realistic representation of NOy descent from the mesophere and thermosphere.

Line 150: NOy in situ production in the mesosphere can be treated in the model.

It could, but as MIPAS measures up to 68 km only, the MIPAS-based parameterization implicitly considers both transport above the mesopause, and production within the mesosphere. Considering mesospheric production on top of the parameterization might lead to double counting of the mesospheric precipitation. It should also be pointed out that this parameterization is recommended for models with their top around the mesopause as part of the solar forcing for CMIP6 and CMIP7 (Matthes et al.,

2017; Funke et al., 2024, see also solarisheppa.kit.edu), and the aim here is to provide solar forcing in ICON in a cost-efficient way as recommended for CMIP6 and CMIP7.

Line 155: I do not understand what was done.

We revised to; Adjustments for solar UV variability (see Section 4.3 for details): The Linoz tables were recalculated for ozone to account for changes in solar UV, particularly in the J-O2 photolysis rates. We move the sentence about J-NO rates extended to the mesosphere (from EMAC) to Section 4.2 which will avoid confusion.

Section 3, last para. I think we should wait for these necessary steps before publication.

We don't understand what this refers to at all. These model experiments were carried out, and results are shown in the paper. we would like to ask the reviewer to clarify what the comment refer to?

Line 164: 2.5-year spin-up time is too short.

What is this statement based on? We tested the spin-up of T and O3 before we made the figures, and 2.5 years is enough. The dynamics in ICON relaxes within a few months, and as the LINOZ ozone only depends on T, it also relaxes very quickly.

Line 168-169: If the ozone is not used for radiation calculation, how can the solar influence be estimated?

Here we aim at the impact of solar forcing on the composition only. There are two pathways which are implicitly considered in the model: the ozone loss due to the EPP indirect effect, and ozone formation due to an increase in short-wave radiation via  $O2 + hv \rightarrow O + O$ ;  $O + O2 \rightarrow O3$ . If the ozone is also used interactively, this chemical impact becomes much more difficult to assess, because there will then be two processes acting on ozone which might counteract each other:  $O3+hv \rightarrow radiative$  heating  $\rightarrow$  higher  $T \rightarrow slower$  reaction O + O2, since this is T-dependent favoring low temperatures. Disentangling this can become very difficult very quickly, and we wanted to focus on the purely chemical response here.

Line 169: What does it mean "Polar chemistry was activated ...". The ozone layer is not prescribed then

This means "Polar spring-time stratospheric ozone loss as seen in the Antarctic ozone hole was activated using the ICON-ART-LINOZ subroutine called PolarChem described in Haenel et al., 2022."

Line 173: Three different scenarios (Stenchikov et al.,1998, 2004, 2009). I do not think 1998 work can be used for 2000-2010 run.

We removed the 1998 citation.

Lines 215-217: What does it mean? It is not understandable w/o missing detailed LINOZ description.

We added more description, please see it above.

Lines 231-234: The explanation is too vague. How Brewer-Dobson circulation (Brewer, 1949) and mesospheric meridional circulation work. In the published papers we do not see robust polar ozone response to solar variability.

This is a misunderstanding – we are not talking about a dynamical feedback here, but about transport of ozone from the source region of largest production in the tropical upper stratosphere and stratopause region, polewards and downwards in the Brewer-Dobson circulation. As we only consider the chemical impact of SSI on ozone, we expect more ozone transport to high latitudes as well in our model experiments. In reality, there is a feedback between dynamical changes and chemical changes which masks or counteracts the purely chemical impact shown here. This should probably be explained in more detail in the paper. We can add one sentence at the end of line 234: "This purely chemical impact in reality could be masked by the feedback between ozone increase and changes in radiative heating, which are not considered here."

Figure 3: The figure is not visible due to overlapping with the legend. There is no doubt that NOy in ICON and EMAC are close because of similar set-up, but what it is not about LINOZ.

The legend can be plotted outside the figure to increase the visibility. As implementing UBCNOy in ICON-ART as a source of EPP NOy acting on (stratospheric) ozone, reproducing NOy at the model top is a necessary prerequisite, and this is demonstrated in Figure 3.

Section 5: Is it about EMAC comparison with ICON-ART or ICON-ART-LINOZ and? It is not clearly stated.

This is about evaluating the changes we made to ICON-ART a) in NOy by implementing UBCNOy, and b) in ozone by connecting LINOZ ozone to NOy, which now includes the UBCNOy compartment. Maybe add a sentence at the beginning of section 5 (before Section 5.1): "In the following, we will evaluate the changes made to ICON-ART. ICON-ART NOy combining with UBCNOy is compared against published model results from EMAC and against MIPAS observations in Section 5.1, the resulting ozone fields and ozone change due

to the additional NOy and solar cycle implementation in LINOZ are discussed in Section 5.2."

Figure 7: There is some agreement, but it is hard to say what the reason for it because the procedure was not clearly accepted. I am surprised that NOy alone (w/o HOx) gives such a strong response in the mesosphere. The ozone response (up to 30-50%) in the stratosphere looks overestimated. From the observations it should be less than 10%.

Figure 7 shows the response of ozone in the stratosphere and mesosphere to implementing UBCNOy in the two models EMAC and ICON. The EMAC results have already been published (Sinnhuber et al 2018), and are used here to evaluate whether the implementation in ICON is consistent. There are three things to point out here: a) EMAC uses MECCA ozone, but does not consider a HOx increase in the mesosphere due to medium-energy electrons in the model experiment shown here. In this sense, the good agreement between ICON, using linearized ozone, with a model using a much more comprehensive chemistry, on mesospheric ozone shows that even in the mesosphere, ICON-ART-LINOZ performs quite well compared to a full chemistry model. b) the stratospheric response depends critically on the downward propagation of the EPP NOy from the source region through the mesosphere to the upper stratosphere. As ICON is freerunning, but EMAC is nudged against meteorological analyses, the year-to-year variability is necessarily different between the two models. This is particularly evident in the Northern hemisphere, where major warmings have a big impact on the indirect effect. Generally the comparison shows that downward transport is more efficient and stable in ICON than in EMAC. This is a feature of the dynamical core of ICON, not of our implementation. It should also be pointed out that this varies greatly from model to model, as can also be seen by the comparison of three models in Sinnhuber et al., 2018. c) MIPAS EPP NOv clearly shows that the indirect effect acts in every winter where observations during polar night exist, see timeseries in Funke et al., 2014 as well as lefthand side of Figure 5. This means that there is an impact on stratospheric ozone in every winter as well. This means that it is not possible to derive the full EPP-driven impact on stratospheric ozone from observations, because there are no stratospheric ozone observations in the high-latitude winter which are not affected by the EPP indirect effect at all. There are some attempts to derive the impact on ozone based on observations (e.g., Fytterer et al., 2014), but these compare ozone from high- and low activity winters. This, by design, has to provide a smaller ozone change than comparing model results with and without the EPP impact. Maybe these three points can be stressed more in the text.

Figure 8: Hard to compare due to different axis. It is strange that there is no

negative ozone response in the mesosphere. Is the Lyman-alpha line missing?

Both figures show comparison of the same model experiments, one showing the full latitude and pressure range, the other focusing only on the mid-to low latitudes and stratosphere for better comparison with observations as shown in Maycock et al (2016). Yes, the absence of a negative ozone response in the mesosphere is likely related to the fact that LINOZ is designed for accurate stratospheric chemistry between 10 and 60 km. Lyman-alpha, which primarily affects J-H2O above ~70 km, is not included, as it has little impact below 70 km.

**References:**

Funke, B., López-Puertas, M., Stiller, G. P., and von Clarmann, T., Mesospheric and stratospheric NOy produced by energetic particle precipitation during 2002–2012, J. Geophys. Res., 119, 4429–4446 (2014), doi:10.1002/2013JD021404.

Fytterer, T., Mlynczak, M. G., Nieder, H., Pérot, K., Sinnhuber, M., Stiller, G., and Urban, J., Energetic particle induced intraseasonal variability of ozone inside the Antarctic polar vortex observed in satellite data, Atmos. Chem. Phys., 15, 3327–3338 (2014), doi:10.5194/acp-15-3327-2015.

Haenel, F., Woiwode, W., Buchmüller, J., Friedl-Vallon, F., Höpfner, M., Johansson, S., Khosrawi, F., Kirner, O., Kleinert, A., Oelhaf, H., Orphal, J., Ruhnke, R., Sinnhuber, B.-M., Ungermann, J., Weimer, M., and Braesicke, P., Challenge of modelling GLORIA observations of upper troposphere–lowermost stratosphere trace gas and cloud distributions at high latitudes: a case study with state-of-the-art models, Atmos. Chem. Phys., 22, 2843–2870 (2022), doi:10.5194/acp-22-2843-2022.

Jungclaus, J. H., Lorenz, S. J., Schmidt, H., Brovkin, V., Brüggemann, N., Chegini, F., et al. (2022). The ICONEarth System Model version 1.0. Journal Advances in Modeling Earth Systems, 14, e2021MS002813. https://doi.org/10.1029/2021MS002813.

Hsu, J., Prather, M. J., Global long-lived chemical modes excited in a 3-D chemistry transport model: Stratospheric N2O, NOy, O3 and CH4 chemistry, Geophys. Res. Lett., 37, L07805 (2010), doi:10.1029/2009GL042243.

Matthes, K., Funke, B., Andersson, M. E., Barnard, L., Beer, J., Charbonneau, P., Clilverd, M. A., Dudok de Wit, T., Haberreiter, M., Hendry, A., Jackman, C. H., Kretzschmar, M., Kruschke, T., Kunze, M., Langematz, U., Marsh, D. R., Maycock, A. C., Misios, S., Rodger, C. J., Scaife, A. A., Seppälä, A., Shangguan, M., Sinnhuber, M., Tourpali, K., Usoskin, I., van de Kamp, M., Verronen, P. T., and Versick, S., Solar forcing for CMIP6 (v3.2), Geosci. Model Dev., 10, 2247–2302 (2017), doi:10.5194/gmd-10-2247-2017.

- Maycock, A. C., Matthes, K., Tegtmeier, S., Thiéblemont, R., and Hood, L., The representation of solar cycle signals in stratospheric ozone Part 1: A comparison of recently updated satellite observations, Atmos. Chem. Phys., 16, 10021–10043 (2016), https://doi.org/10.5194/acp-16-10021-2016.
- McLinden, C. A., Olsen, S. C., Hannegan, B., Wild, O., Prather, M. J., and Sundet, J., Stratospheric ozone in 3-D models: A simple chemistry and the cross-tropopause flux, J. Geophys. Res.-Atmos., 105, 14653–14665 (2000), doi:10.1029/2000JD900124.
- Olsen, SC., McLinden, CA, and Prather, MJ. Stratospheric N2O-NOy system, Testing uncertainties in a three-dimensional framework. Journal of Geophysical Research Atmospheres, 106(D22), 28771-28784 (2001), doi:10.1029/2001JD000559.
- Prather, M.J., J. Hsu, N.M. DeLuca, C.H. Jackman, L.D. Oman, A.R. Douglass, E.L. Fleming, S.E. Strahan, S.D. Steenrod, O.A. Søvde, I.S.A. Isaksen, L. Froidevaux, and B. Funke (2015) Measuring and modeling the lifetime of nitrous oxide including its variability, J. Geophys. Res. Atmos., 120, 5693–5705. doi: 10.1002/2015JD023267.
- Raschendorfer, M., 2001: The new turbulence parameterization of LM. COSMO News Letter No. 1, Consortium for Small-Scale Modelling, 89–97, URL http://www.cosmo-model.org.
- Rieger, D., Bangert, M., Bischoff-Gauss, I., Förstner, J., Lundgren, K., Reinert, D., Schröter, J., Vogel, H., Zängl, G., Ruhnke, R., and Vogel, B., ICON–ART 1.0 a new online-coupled model system from the global to regional scale, Geosci. Model Dev., 8, 1659–1676 (2015),doi:10.5194/gmd-8-1659-2015.
- Ruiz, D. J. and M.J. Prather (2022) From the middle stratosphere to the surface, using nitrous oxide to constrain the stratosphere–troposphere exchange of ozone, Atmos. Chem. Phys., 22, 2079–2093, doi: 10.5194/acp-22-2079-2022.
- Sander, R., Baumgaertner, A., Gromov, S., Harder, H., Jöckel, P., Kerkweg, A., Kubistin, D., Regelin, E., Riede, H., Sandu, A., Taraborrelli, D., Tost, H., and Xie, Z.-Q.: The atmospheric chemistry box model CAABA/MECCA-3.0 (2011), Geosci. Model Dev., 4, 373–380, https://doi.org/10.5194/gmd-4-373-2011.
- Sandu, A. and Sander, R.: Technical note: Simulating chemical systems in Fortran90 and Matlab with the Kinetic PreProcessor KPP-2.1, (2006), Atmos. Chem. Phys., 6, 187–195, https://doi.org/10.5194/acp-6-187-2006.
- Sandu, A., Verwer, J., Blom, J., Spee, E., Carmichael, G., and Potra, F.: Benchmarking stiff ode solvers for atmospheric chemistry problems II: Rosenbrock solvers, (1997), Atmos. Environ., 31, 3459–3472, https://doi.org/10.1016/S1352-2310(97)83212-8.
- Seppälä, A., Kalakoski, N., Verronen, P.T. *et al.* Polar mesospheric ozone loss initiates downward coupling of solar signal in the Northern Hemisphere. *Nat Commun* **16**, 748 (2025). https://doi.org/10.1038/s41467-025-55966-z.
- Schröter, J., Rieger, D., Stassen, C., Vogel, H., Weimer, M., Werchner, S., Förstner, J., Prill, F., Reinert, D., Zängl, G., Giorgetta, M., Ruhnke, R., Vogel, B., and Braesicke, P., ICON-ART 2.1: a flexible tracer framework and its application for composition studies in

numerical weather forecasting and climate simulations, Geosci. Model Dev., 11, 4043–4068 (2018), doi:10.5194/gmd-11-4043-2018.

Sinnhuber, M., Berger, U., Funke, B., Nieder, H., Reddmann, T., Stiller, G., Versick, S., von Clarmann, T., and Wissing, J. M., NOy production, ozone loss and changes in net radiative heating due to energetic particle precipitation in 2002–2010, Atmos. Chem. Phys., 18, 1115–1147 (2018), doi:10.5194/acp-18-1115-2018.

Stevens, B., Giorgetta, M., Esch, M., Mauritsen, T., Crueger, T., Rast, S., et al. (2013). Atmospheric component of the MPI-M earth system model: ECHAM6. Journal of Advances in Modeling Earth Systems, 5, 146–172. https://doi.org/10.1002/jame.20015.

Weimer, M. Towards Seamless Simulations of Polar Stratospheric Clouds and Ozone in the Polar Stratosphere with ICON-ART. PhD Thesis. (2019), Germany: Karlsruhe Institute of Technology (KIT).

Zängl, G., Reinert, D., and Prill, F.: Grid refinement in ICON v2.6.4, Geosci. Model Dev., 15, 7153–7176 (2022), doi:10.5194/gmd-15-7153-2022.

Zängl, G., D. Reinert, P. Rípodas, and M. Baldauf, 2015: The ICON (ICOsahedral Non-hydrostatic) 770 modelling framework of DWD and MPI-M: Description of the non-hydrostatic dynamical core. 771 Q J R Meteorol Soc, 141 (687), 563–579, https://doi.org/https://doi.org/10.1002/qj.2378.

ICON Tutorial, working with the ICON Model F. Prill, D. Reinert, D. Rieger, G. Z"angl, 2020.